# Distinct place cell dynamics in CA1 and CA3 encode experience in new environments

Can Dong [1], Antoine D. Madar[1] & Mark E. J. Sheffield [1✉]

When exploring new environments animals form spatial memories that are updated with experience and retrieved upon re-exposure to the same environment. The hippocampus is thought to support these memory processes, but how this is achieved by different subnetworks such as CA1 and CA3 remains unclear. To understand how hippocampal spatial representations emerge and evolve during familiarization, we performed 2-photon calcium imaging in mice running in new virtual environments and compared the trial-to-trial dynamics of place cells in CA1 and CA3 over days. We find that place fields in CA1 emerge rapidly but tend to shift backwards from trial-to-trial and remap upon re-exposure to the environment a day later. In contrast, place fields in CA3 emerge gradually but show more stable trial-to-trial and day-to-day dynamics. These results reflect different roles in CA1 and CA3 in spatial memory processing during familiarization to new environments and constrain the potential mechanisms that support them.

[1] Department of Neurobiology and Institute for Neuroscience, University of Chicago, Chicago, IL, USA. ✉email: sheffield@uchicago.edu

The hippocampus has a critical role in episodic memory by rapidly forming, updating, and retrieving patterns of activity that represent specific memories[1,2]. The CA1 subnetwork is considered the main output of the hippocampus that transmits information to the cortex and other regions[3], but the computational role of CA1 in memory processing remains unclear. Theoretical studies generally highlight networks upstream of CA1, such as CA3 and the medial entorhinal cortex as being attractor networks that encode and retrieve representations associated with different environments[3–7]. These representations could then simply be inherited by CA1[8–10]. However, given the many forms of synaptic plasticity at CA1 synapses[11,12], the complex dendritic computations performed by CA1 neurons[13–15] and the diversity of CA1 interneurons[16,17], CA1 activity dynamics are unlikely to be purely inherited from upstream regions. Establishing the difference in activity dynamics between CA1 and its inputs will help reveal how information is processed by CA1, which computations are specific to CA1, and what the role of its inputs are.

Spatial memories are thought to be encoded and retrieved in the hippocampus through the activity of place cells[18–22]—cells with spatially selective firing fields called place fields (PFs). All hippocampal subnetworks (CA1, CA2, CA3, and dentate gyrus) express PFs during navigation[10,18]. Importantly, CA1 pyramidal neurons develop PFs rapidly during the exploration of a novel environment[23]. Understanding how hippocampal representations emerge so rapidly during one-shot events is critical to refine theories of memory. However, the specific dynamics and underlying mechanisms of PF emergence in CA1 have only recently come into focus, and even less is known about PF emergence in CA3, the main input region to CA1[11,14,15,24–27]. An important step forward came from analyzing PFs on a trial-to-trial basis during the very first moments in a novel environment[15]. Such trial-to-trial resolution showed that many neurons in CA1 develop a PF on the first trial and engage synaptic plasticity mechanisms in the form of increased dendritic branch spike prevalence[14,15]. Determining the dynamics of PF emergence in CA3 under the same conditions in novel environments will generate new insights into the mechanisms of PF emergence in CA1 and reveal the extent to which CA1 PF emergence is inherited from CA3 PFs.

In addition to how spatial representations emerge, understanding how they evolve during familiarization to a novel environment (a form of spatial learning) is also critical to refine theories of learning and memory. Little is known about trial-to-trial PF dynamics in CA1 or CA3 with most studies focusing on mean PF dynamics across conditions[18,28,29]. In familiar environments, it has been shown that PFs in both CA1 and CA3 tend to shift backwards with experience and develop negatively skewed PFs to varying degrees depending on the behavioral paradigm used[29–32]. This is thought to be an experience-dependent process that requires NMDA receptor-dependent long-term synaptic plasticity[33]. How soon these phenomena appear during the familiarization process, and whether they are inherited by CA1 from CA3, remains unclear[31]. Tracking large numbers of CA1 and CA3 neurons under the same conditions during familiarization to novel environments will help better understand these phenomena.

Memory retrieval is thought to be achieved by reinstating the same neuronal activity that occurred during learning[10,34,35]. However, evidence from recordings of large ensembles of place cells have shown that many PFs in CA1 are unstable across exposures to the same familiar environment[13,18,28,36–39]. Again, as with PF emergence and trial-to-trial dynamics, a lot less is known about the stability of CA1 and CA3 PFs during re-exposure to a novel environment. This has left unclear how

familiarization influences PF stability. CA1 PFs might be expected to be less stable across days than CA3 PFs as CA1 has been shown to integrate stable spatial information from CA3 with time signals from CA2 upon re-exposures to the same familiar environment[10]. Further supporting the idea of reactivation of stable PFs in CA3 comes from its hypothesized role in pattern completion[7], although recent experimental findings tracking PFs in novel environments across days suggest otherwise[18]. Lastly, trial-to-trial retrieval dynamics of single PFs during re-exposure to a novel environment are not known because of the technical difficulty in tracking the same place cells across days[31]. Overall, how PFs emerge, evolve, and stabilize during familiarization to a novel environment is unclear, both in CA1 and CA3.

In this work, we use two-photon $Ca^{2+}$ imaging to longitudinally record from large populations of CA1 and CA3 pyramidal neurons in head-fixed mice running unidirectionally on a treadmill to repeatedly traverse visually enriched virtual linear environments with consistent behavior. We track neurons from the very first moments in a novel environment and across days to compare the emergence and ongoing dynamics of PFs. We find that PFs initially emerge much faster in CA1 than CA3, but CA1 PFs on average continuously shift backwards with experience. CA3 PFs emerge relatively slowly but are subsequently more reliable than CA1 PFs across trials, displaying less backward shifting. We also find that PF backward shifting decreases with familiarization across trials and across days. Upon re-exposure to the novel environment on the second day, stable PFs in CA3 reactivate rapidly on the first trial, whereas CA1 PFs demonstrate a higher propensity to remap across days. Our findings demonstrate major differences in the initial emergence, shifting, and stability of PFs in CA1 and CA3 during familiarization to a novel environment. The distinct features of CA1 PFs compared with CA3 PFs suggest that the CA1 performs significant computations on its spatial inputs from CA3 to support a distinct role in spatial memory processing.

## Results

**PF emergence in a novel environment in CA1 and CA3.** We expressed GCaMP6f in either dorsal CA1 or CA3a (referred to as CA3 from here on) of different mice (Fig. 1c, d). The Grik4-cre line[40] was used to restrict expression to CA3 pyramidal neurons (Fig. 1c). Importantly, these mice show CRE expression in ~100% of pyramidal cells in CA3, which means our recordings were not biased to a sub-population of CA3 pyramidal cells[40]. Using two-photon microscopy we then recorded calcium transients from pyramidal cell populations in both regions (Supplementary Fig. 1)[13,15]. On experimental day 1 mice were exposed to a familiar (F) environment before being switched to a novel environment (N1) (Fig. 1a)[15]. On experimental day 2, mice experienced the same F-to-N switch but to a different N environment (N2; Fig. 1a). N1 and N2 were grouped together and are referred to as N. Mice momentarily slowed down after the transition between environments (Fig. 1b), confirming their perception of the switch to N. Because mice were restricted to running in 1 dimension on a custom-built treadmill, this paradigm led to many repeated traversals in both environments with matched behavior, allowing lap-by-lap PF dynamics to be measured systematically and compared across F and N environments without confounds caused by changes in behavior.

As has previously been reported, both CA1 and CA3 place cells globally remapped upon exposure to N (Fig. 2a, b) and displayed altered PF properties compared with F (Supplementary Fig. 2)[4,6,15,18,41–44]. For instance, PF widths were on average larger and PF precision lower in N than F in both CA1 and CA3. Out-of-field PF firing also increased in N versus F as did PF

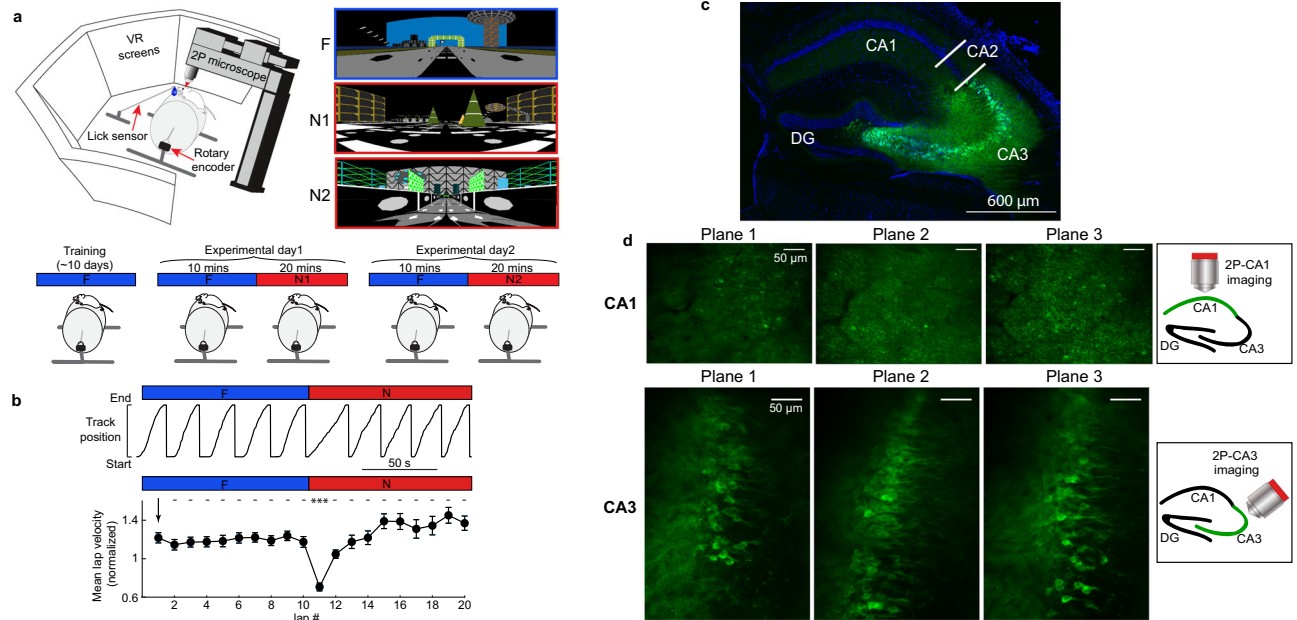

**Fig. 1 Experimental setup. a** Top left, depiction of the virtual reality (VR) setup. Top right, the familiar (F) and two novel (N1 and N2) environments. Bottom: scheme of the experimental procedure. **b** Top: single mouse behavior showing track position versus time during an F to N switch. Bottom: summary data across all mice of mean lap velocity during F and N (*n* = 20 sessions in 11 mice). Lap velocity is normalized to the mean lap velocity in each mouse ±SEM, and each lap is compared to the first lap in F using a one-way ANOVA with Tukey HSD post hoc test. ***P < 0.001, P = 6.3 × 10$^{-15}$. **c** Brain slice showing specific CA3 expression of cre-dependent GCaMP6f (green) and DAPI for nuclei (blue) from a Grik4-cre transgenic mouse. **d** Example field of views (FOV) from multiplane imaging in CA1 (top) and CA3 (bottom). Right: scheme of the position for the objective during imaging.

transient occupancy. Crucially, we wanted to observe the real-time emergence dynamics of new place cells in CA1 and CA3 to examine potential differences. Therefore, the emergence of new PFs in N was quantified on a lap-by-lap basis (Fig. 2c–e). Some PFs formed instantly, i.e., on the first lap (instant PFs; Fig. 2c; left), whereas others were delayed by several laps (delayed PFs; Fig. 2c; right). Similar to previous observations, many CA1 PFs formed rapidly[14,15,23], with a high proportion of instant PFs (30%; Fig. 2d, e). Unexpectedly, only 9% of CA3 PFs were instant and the distribution of PF onset lap number was more uniform, indicating CA3 PFs form more gradually than CA1 PFs (Fig. 2d, e; Supplementary Fig. 2c, d). Supporting this, CA1 place cell activity decoded position on the first lap better than CA3 place cell activity (Fig. 2f–h, Supplementary Fig. 3). These data suggest that in a novel environment, CA1 instantly forms a well-organized map, whereas CA3 forms a map gradually with experience.

**Trial-to-trial PF dynamics in a novel environment.** To examine and compare how PFs evolve in CA1 and CA3 during spatial learning, we tracked new PFs throughout familiarization to N. We compared the first and second half of the session and found CA3 PFs were more stable than CA1 PFs within a session in N (Fig. 3a). Next, we computed each PF's center of mass (COM) on a lap-to-lap basis (Fig. 3b): in CA1, approximately half of the PFs significantly shifted during the session, but only a third in CA3 (Fig. 3c). The direction of shift could be backward or forward relative to the direction of the animal's motion (Fig. 3b–d) but most PFs shifted backwards, with a larger skew in CA1 than CA3 (Fig. 3c–d). As a population, CA1 PFs shifted backwards much faster than CA3 PFs (Fig. 3e). This difference is not an artifact of a lower sample size of CA3 PFs as shown by downsampling 1000 times the CA1 data set (Fig. 3e). Removing transients prior to the onset of a robust PF (as we do for defining PF onset lap, see methods) did not alter our conclusions (Supplementary Fig. 4a–b). The shift of individual PFs calculated across all laps

within the session was not related to PF onset lap or velocity (Supplementary Figs. 5 and 6). Backward and forward shifting PFs occurred at all positions, with large shifts weakly biased towards the end of the track for CA1 (Supplementary Fig. 7).

Backward shifting PFs have been previously reported and may be accompanied by an increase in PF width and the development of a negative skew with experience, although reports are conflicting[29–33,45]. We found that, in a novel environment, PF width tended to increase through the first 10 laps in CA1 but not in CA3 (Supplementary Fig. 8). PFs also started with a positive skew that decreased with experience in CA1 but not in CA3 (Supplementary Fig. 8). Note that under calcium imaging, as compared with electrophysiology, PFs are generally positively skewed rather than symmetric as an artifact of calcium transient decay times[46]. An evolution towards less positively skewed yet larger PFs could explain backward shifting of PF COMs (see example in Supplementary Fig. 8a), consistent with some electrophysiological reports[32,45]. However, changes in skewness and width were not the only cause of COM backward shifting, as the PFs end position also shifted backwards (Supplementary Fig. 4c–d).

Interestingly, we noticed that in CA1 the population tended to shift forward during the first laps, with the global backward shift occurring around the fifth lap (Fig. 3e). We found that the delay in backward shifting was driven by instant PFs that tended to shift forward on early laps, even though their overall shift calculated across the entire session was backward (Supplementary Fig. 9). It is worth restating here that CA3 shows few early-onset PFs on the initial laps which is when the forward shifting of instant CA1 PFs occurs. This suggests backward shifting in CA1 might require the presence of established CA3 PFs, which is why instant PFs initially shift forward.

**PF stability upon re-exposure to a novel environment across days in CA1 and CA3.** Memory recall of spatial environments is

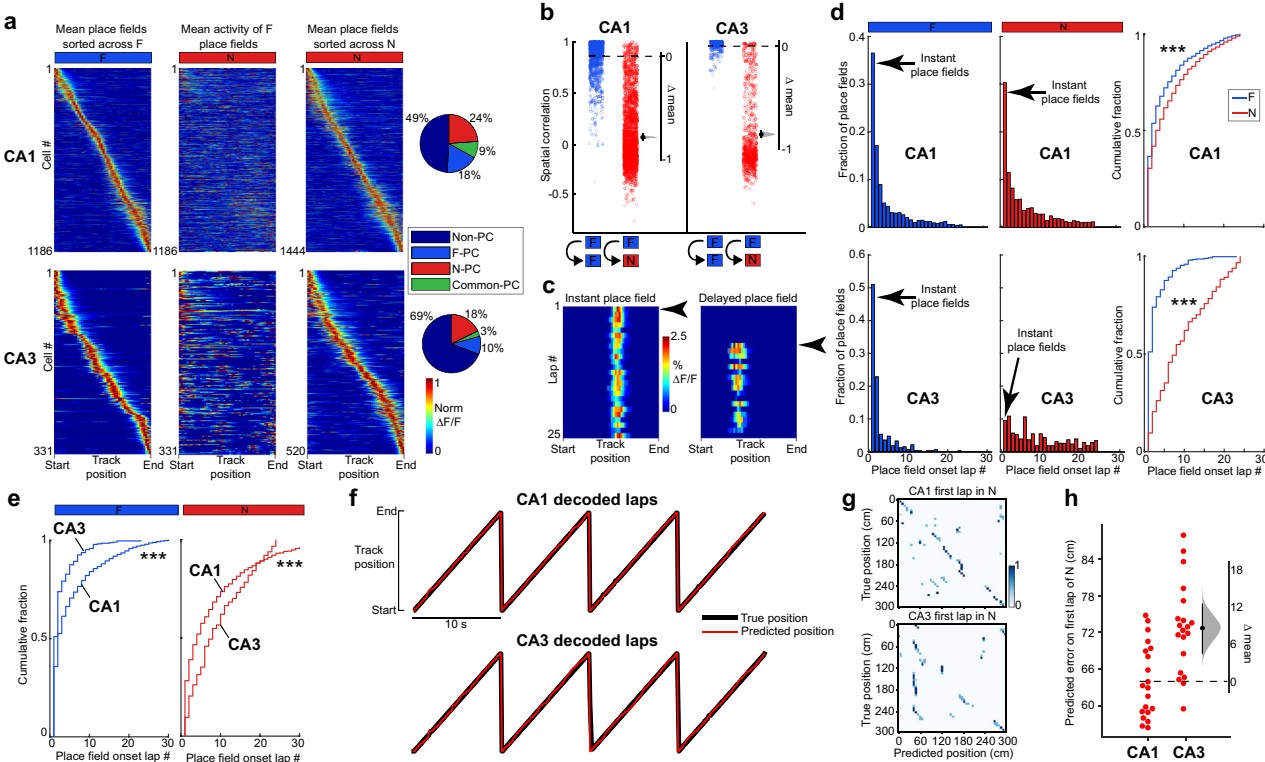

**Fig. 2 Place field emergence in novel environments is rapid in CA1 but gradual in CA3. a** Left: mean place fields (PFs) in F sorted by track position. Middle: mean activity of the same neurons on the left in N. Right: mean place fields in N sorted by track position. Far right: percentage of place cells in F and N or common to both from all active neurons. CA1, top panels; CA3, bottom panels. ΔF/F activity is normalized to each neurons' maximum transient amplitude. $n = 7$ sessions in four mice for CA1, $n = 13$ sessions in seven mice for CA3. Only place cells with single PFs are displayed, but place cells with multiple PFs are included in the pie charts and later analyses. **b** Pearson's correlation coefficient of each cell's average activity map within F (blue, CA1 $n = 1392$, CA3 $n = 320$ neurons with activity in both halves of the F session) and between F and N (red, CA1 $n = 2283$, CA3 $n = 698$ neurons with activity in both sessions and a PF in one of them). Bootstrapped mean difference (Δ) with the distribution of the mean (gray) and 95% confidence interval (error bar) on the right of each plot (see Methods). **c** Example of an instant PF (left) and delayed PF (right) in N. Arrows indicate PF onset lap. **d** Histograms of PF onset laps in F (left, CA1 $n = 1643$ PFs, CA3 $n = 336$ PFs) and N (middle, CA1 $n = 1712$ PFs, CA3 $n = 532$ PFs) and cumulative fraction plots for F and N (right). Wilcoxon rank-sum test, two-sided. ***$P < 0.001$, CA1, $P = 3.9 \times 10^{-18}$, CA3, $P = 1.6 \times 10^{-55}$. **e** Cumulative fraction plots for CA1 and CA3 PF onset lap in F (left) and in N (right). Wilcoxon rank-sum test, two-sided. ***$P < 0.001$, F, $P = 2.9 \times 10^{-13}$, N, $P = 7.4 \times 10^{-13}$. **f** Example mouse showing true track position (black) on laps 37–40 and the predicted position (red) decoded by an LSTM decoder (see Methods). CA1, top; CA3, bottom. **g** Confusion matrix between the predicted ($x$ axis) and the true ($y$ axis) position for the first lap in N. **h** Predicted error on the first lap in N for CA1 and CA3. Each dot represents the decoding error for one decoder trial built based on the activity of 200 randomly chosen place cells from CA1 (left) or CA3 (right). $n = 20$ decoder trials. Bootstrapped mean difference (Δ) with 95% CI (error bar) on the right.

generally thought to be supported by the reactivation of stable PFs upon re-exposures[19–21,47]. We, therefore, examined the same place cells upon re-exposure to N (specifically N2, see methods) across days (Fig. 4a). On N day 2, increased lap velocity on the first lap compared to the first lap on N day 1 revealed mice had become more familiar with N (Fig. 4b). We then quantified the spatial correlation of mean PFs identified on N day 1 with N day 2, which was on average significantly higher in CA3 (0.70 ± 0.04) than CA1 (0.49 ± 0.02) (Fig. 4c–f). The bimodal distribution of CA3 PF spatial correlations (Fig. 4d; bottom) helped categorize PFs as either stable ($R > 0.5$) or unstable ($\leq 0.5$). In all mice, we found that CA3 had a higher fraction of stable PFs compared to CA1 (Fig. 4e). Interestingly, in CA1, we found a small but significant positive correlation between PF shifting on N day 1 and PF spatial correlation across days, suggesting that higher day-to-day PF stability is associated with more stable trial-to-trial PF dynamics upon the first exposure to a novel environment (Supplementary Fig. 10).

We then compared the PF onset laps of stable PFs on day 2 (re-emergence, Fig. 4g, left) with PFs that newly formed on day 2, which included both unstable PFs from day 1 (Fig. 4g, middle)

and PFs that appeared for the first time on day 2 (Fig. 4g, right). We found that stable PFs re-emerged earlier in the session than newly formed PFs, and this difference was much more apparent in CA3 than CA1 (Fig. 4h, right). Further, a high proportion of CA1 PFs is continuously forming even as the environment becomes familiar. Lastly, similar to day 1, the vast majority of new PFs in CA3 emerge gradually after a delay with very few instant PFs (Fig. 4h, bottom–middle and bottom–right) whereas the CA1 again shows a much higher proportion of instant PFs on day 2 (Fig. 4h, top-middle and top-right). This reveals that stable CA3 PFs are retrieved much faster than new PFs emerge during re-exposure on day 2, whereas stable and new PFs in CA1 show similar emergence dynamics.

**Lap-by-lap dynamics across days.** We first asked if there was any evidence of continuous PF shifting of the population occurring offline between sessions separated by a day. We found no such evidence as the distribution of PF shift distance between the end of the session on day 1 and the start of the session on day 2 was not skewed (Fig. 5a). This indicates that COM shifting reflects

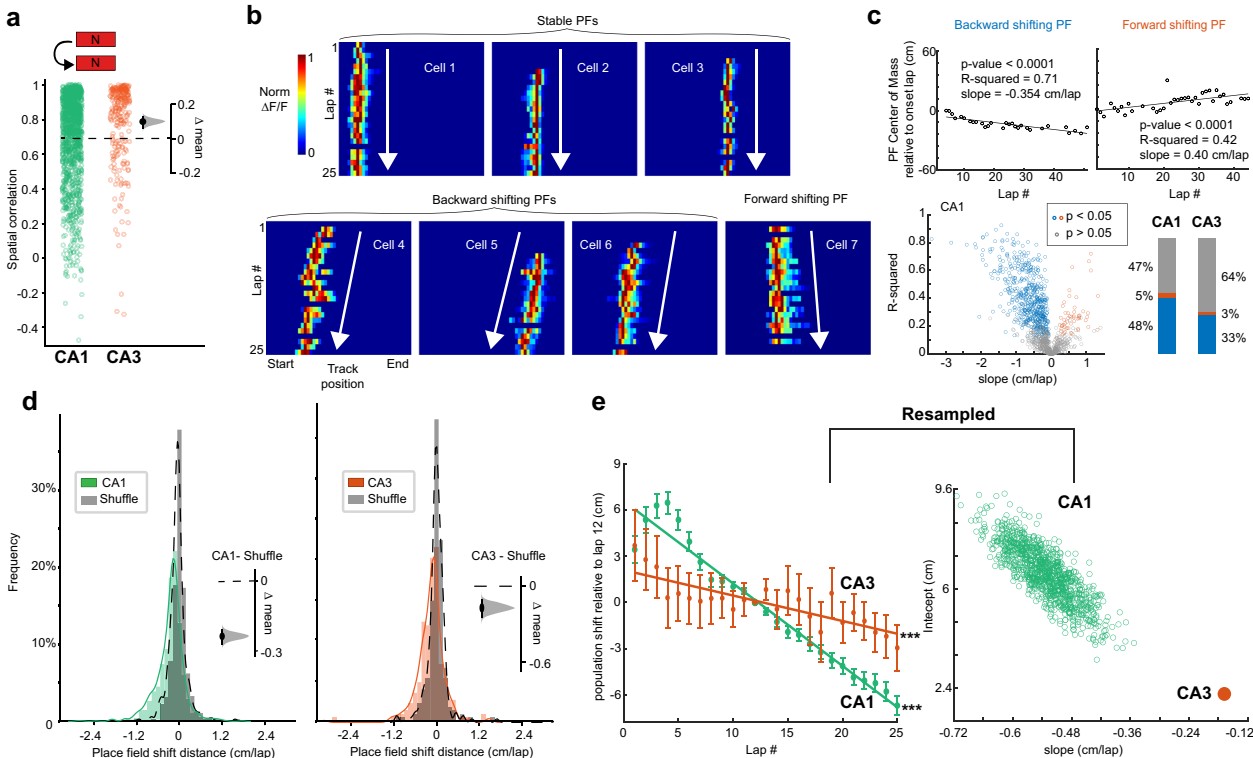

**Fig. 3 CA3 place fields exhibit relative lap-by-lap stability while CA1 place fields shift with experience in novel environments. a** Pearson's correlation coefficient of all neurons with place fields within N (average activity map in first 10 laps versus next 10 laps of session: CA1 $n = 1238$, CA3 $n = 255$ neurons). Bootstrapped mean difference ($\Delta$) between CA1 and CA3 shown on the right. **b** Example place cells with stable or shifting place fields (PFs) in N (lap-by-lap activity for the first 25 laps). White arrows depict the direction of shifting. **c**, Linear regression analysis on the center of mass (COM) of individual PFs (CA1 in N) shows a high number of significantly shifting PFs (blue and red dots and bars). The slope measures the direction and amplitude of the PF shift: for significantly shifting PFs, backward is blue, forward is red. CA3 has a lower proportion of significantly shifting PFs than CA1. **d** Histograms of the COM difference between last and onset laps (five-lap weighted average, see Methods) normalized by the number of laps for each PF in N for CA1 and CA3. Shuffling the lap numbers (gray distributions) reveals an overrepresentation of backward shifting PFs, as confirmed by the bootstrapped mean difference ($\Delta$, insets). **e** Population shift of COM. Left, mean ±SEM, over all PFs, of five-lap sliding average COM difference relative to lap 12 (see Methods), in N. CA1 (green) and CA3 (orange). Linear regression analysis (on all data points, not means) shows that the population of PFs shifts backwards significantly, both in CA1 and CA3. Linear regression, $F$ test, ***$P < 0.001$, CA1, $P < 1 \times 10^{-100}$, CA3, $P = 1.3 \times 10^{-4}$. Right, The CA1 data set was resampled 1000x using $n = 175$ PFs to match the number of CA3 PFs and the slope and intercept of the regression line were measured each time (green dots). CA1 slopes are always steeper than the CA3 slope indicating that the CA1 population shifts significantly faster than CA3.

experience-dependent processes that occur during ongoing familiarization but do not continue offline. However, when considering only stable PFs that on day 1 showed statistically significant backward shifting, we found they tended to reset on day 2 toward their original position on the early laps of day 1 (Fig. 5b, Supplementary Fig. 11). Therefore, although shifting does not continue offline, offline processes may be involved in resetting PFs back towards their original position.

We then asked whether the PF shifting dynamics we had observed on day 1 in N continued upon re-exposure on day 2. The few stable PFs that exhibited significant shifting on both days did not change much on average, but the direction and amplitude of individual PF shifts were not correlated across days (Fig. 5c). At the population level, which was thus mostly driven by newly shifting PFs, we found that CA1 PFs on average shifted backwards like on day 1, before stabilizing after lap 15 (Fig. 5d). The CA3 map shifted less than CA1 (Fig. 5d, right). In contrast to day 1, we observed a decrease in PF width over the first laps in CA1, but not CA3, and little change in skewness with population dynamics indistinguishable between CA1 and CA3 (Supplementary Fig. 8).

Formally comparing dynamics on N day 1, N day 2, and very familiar environments (F), we found that CA1 population

backward shifting slowed down and stabilized with familiarization across days (Fig. 6a, c). Stabilization also tended to occur earlier with the level of familiarity: in contrast to day 1, a plateau was reached on day 2 after 15 laps, and only after 12 laps in F. Shifting of the CA3 map showed a similar trend than CA1 although changes with familiarization across days were less obvious (Fig. 6b, d), in part because shifting dynamics were already slow on day 1. Resampling the CA3 data revealed extensive overlap in the population shift across conditions, indicating similar dynamics, (Fig. 6d, right). This analysis also revealed that the population becomes more homogeneous with familiarization across days (smaller variance in the slope distribution). Finally, in CA1, PF skewness, and width tended to evolve and stabilize with familiarization, whereas the CA3 population did not show clear dynamics despite homogenizing across days (Supplementary Fig. 8). These results suggest that lap-wise PF dynamics are enhanced by novelty and decrease as a function of familiarity in CA1, and these PF changes are less apparent in CA3.

## Discussion
We used two-photon calcium imaging of large populations of dorsal CA1 and CA3 pyramidal neurons to measure and compare the

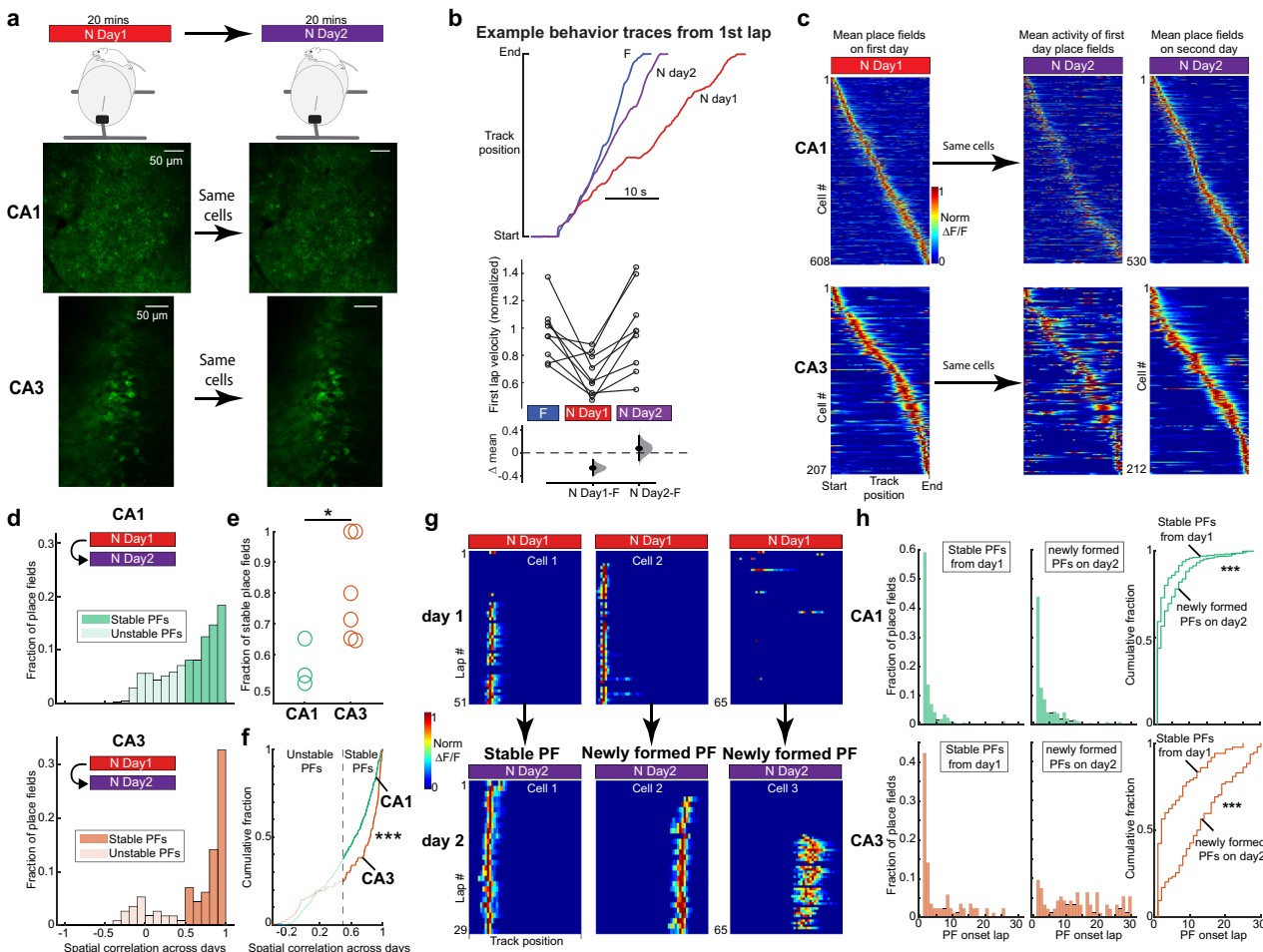

**Fig. 4 CA3 place fields exhibit higher stability across days than CA1 place fields and rapidly reappear upon re-exposure to the environment. a** Experimental setup. Top: mice are recorded for 20 min in N across 2 days. Bottom: example field of view from one imaging plane from CA1 and CA3 across 2 days showing the same cells. scale bar = 50 μm. **b** Top, example mouse behavior showing track position for the first lap in F (blue), N day 1 (red), and N day 2 (purple). Bottom, summary across all mice of first lap velocity in F, N day 1, and N day 2. Lap velocities were normalized to the mean velocity in F in each mouse (n = 11 mice). Bootstrapped mean difference (Δ) between F and N day 1, and F and N day 2. Note that lap velocity is only different on N day 1. **c** Left: mean place fields in N on day 1 sorted by track position. Middle: mean activity of the same neurons on the left on day 2 in N. Right: mean place fields in N on day 2 sorted by track position. **d** Histograms of Pearson's correlation coefficient of average activity map across days in N. Spatial correlations >0.5 were considered stable place fields and less than 0.5 unstable. CA1 n = 409, CA3 n = 113 neurons with PFs in both sessions. **e** Fraction of stable place fields across days per mouse in CA1 and CA3. Wilcoxon rank-sum test, two-sided. *P < 0.05, P = 0.0476. **f** Cumulative fraction plots of Pearson's correlation coefficient in N across days in CA1 and CA3 from the same data shown in **c**. Wilcoxon rank-sum test, two-sided. ***P < 0.001, P = 6.9 × 10$^{-4}$. **g** Example of a stable place field (left), an unstable place field with a newly formed place field on day 2 (middle), and a newly formed place field on day 2 (right). Place field transients showed lap-by-lap for all laps in N on day 1 and day 2 from the same three cells. **h** Histograms of place field onset laps in N on day 2 for stable place fields (left), and newly formed place fields (middle) in CA1 (top) and CA3 (bottom). Right: cumulative fraction plots of histogram data. Wilcoxon rank-sum test, two-sided. ***P < 0.001, CA1, P = 1.8 × 10$^{-5}$, CA3, P = 6.7 × 10$^{-10}$. CA1 stable n = 251 PFs, newly formed, n = 333 PFs, CA3 stable n = 72 PFs, newly formed, n = 94 PFs.

emergence, shifting dynamics, and longitudinal stability of PFs in novel environments with trial-to-trial resolution. We found that PFs emerge faster in CA1 but place cells are constantly renewed across exposures, whereas they emerge later in CA3 with less turnover across exposures. After emergence, the location of the PFs is not always stable, sometimes showing prominent backward or forward shifting from lap-to-lap. The average spatial representation in the hippocampus shifts backwards, with a faster shift in CA1 than CA3. This backward shifting slows down with familiarization across days. These findings support the idea that CA3 and CA1 perform distinct functions during familiarization to a novel environment, and CA1 does not simply inherit spatial information from CA3 during this form of spatial learning. They also constrain the potential mechanisms explaining how spatial representations emerge, evolve, and reactivate in the hippocampus.

Consistent with previous reports, we observed hippocampal PFs that emerged instantly in a novel environment (on the very first trial) and others that emerged after multiple trials (delayed-onset PFs)[14,15,23]. Interestingly, the proportion of instant PFs was much lower in CA3 than CA1 (Fig. 2). Although to our knowledge this is the first time the trial-to-trial emergence of CA3 PFs has been reported in completely novel environments, a related study did compare the emergence of PFs in CA3 and CA1 in a familiar linear belt with novel sensory cues added[29]. This study found two types of place cells: (1) landmark vector (LV) cells that have multiple instant PFs locked to sensory cues and (2) place cells with single PFs with a mix of instant and delayed-onset PFs that emerge when novel sensory cues are added. Interestingly, they found 10 times more LV cells in CA1 than CA3. Because PFs in LV cells emerged instantly, the larger proportion of LV cells in

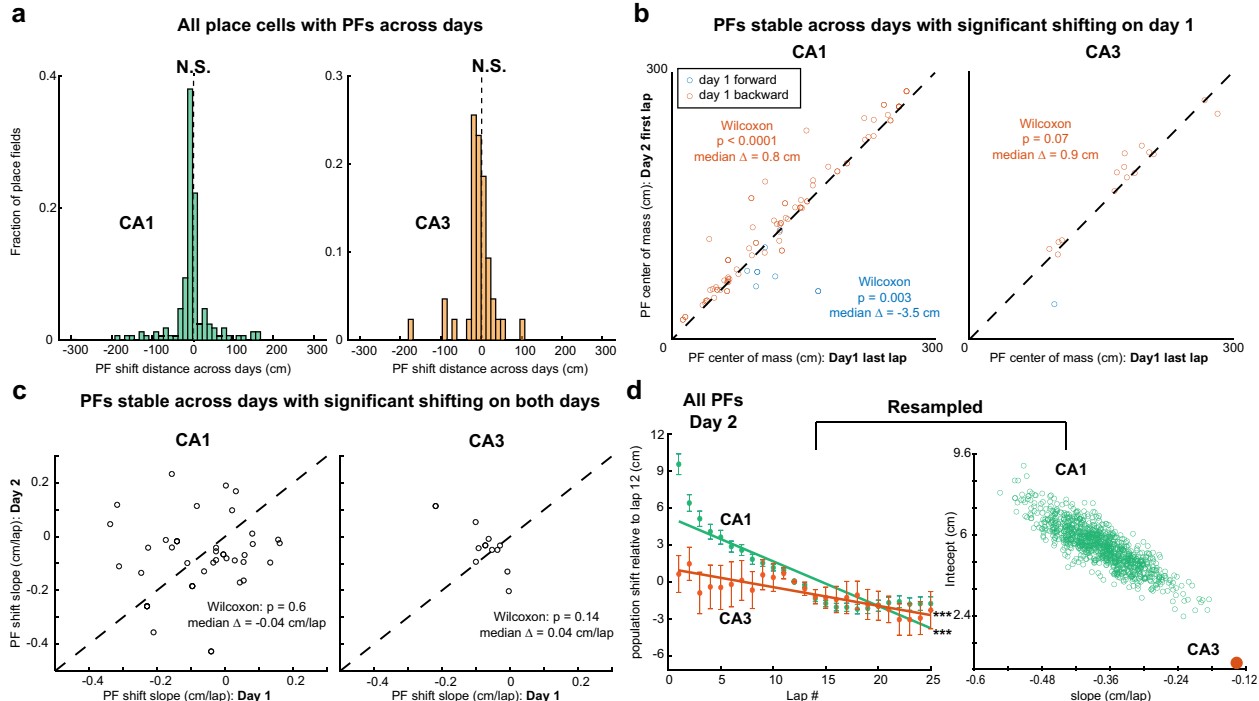

**Fig. 5 Place field shifting resets and continues upon first re-exposure to N on day 2. a** Histogram of place field (PF) shifts across days for CA1 (left, $n =$ 171 PFs) and CA3 (right, $n = 43$ PFs). The shifts were calculated as the difference between each PF's center of mass (COM) on the last active lap on day 1 and first active lap in day 2. Only place cells with PFs on both days are included. Wilcoxon signed-rank test, two-sided. N.S., $P > 0.05$. **b** Comparison of COM position at the end of day 1 and COM position at the start of day 2 for PFs with day-to-day correlation >0.5 and significant shifting on day 1 (see Fig. 3c). Dashed diagonal: identity line. On day 2, PFs tend to reset towards the direction opposite of shifting on day 1. Two-sided Wilcoxon sign-rank test on Day 2-Day 1 difference from 0 for backward and forward shifting ($P$ values and median in panel). **c** Comparison of PF shifting slope on both days. No clear correlation is observed, but the Day 2-Day 1 difference is not significantly different from 0 (Wilcoxon sign-rank, two-sided: $P$ values and medians in panel) and individual PF shifting was slow (close to 0 cm/lap) on both days. **d** Population shift of PF COM. Left, mean ±SEM, over all PFs, of five-lap sliding average COM difference relative to lap 12 (see Methods), in N day 2. CA1 (green; $n = 371$ PFs from three mice) and CA3 (orange; $n = 84$ PFs from six mice). Linear regression, F test, ***$P < 0.001$ CA1, $P < 1 \times 10^{-100}$, CA3, $P = 1.1 \times 10^{-4}$. The resampling analysis (1000 random resamples matching the CA3 sample size) shows that CA1 slopes are always steeper than the CA3 slope, indicating that on day 2 the CA1 population shifts backward faster than CA3.

CA1 is consistent with our finding that CA1 has more instant PFs than CA3. We confirmed that CA1 has a lot more place cells with multiple PFs than CA3 (10–15% in N and F in CA1 vs 1–2% in CA3), but, in contrast to LV cells, only a fraction of them had instant PFs (31% in CA1 in N, comparable to the 27% instant PFs of single-PF place cells). There was no difference between multi- and single-PF place cells in the distribution of onset laps (two-sample KS-test $p = 0.4$ in N, CA1). This might be due to differences in paradigms. For instance, in contrast to the global remapping and emergence of PFs distributed across the entire track that we see in new environments, adding a cue to a familiar environment only induces partial remapping and PF emergence near cues[29].

Rapid PF emergence in a novel environment in CA1 is thought to rely on a combination of strong, spatially tuned excitatory inputs that do not need to be potentiated[14] and high neuronal excitability, either from novelty-induced disinhibition[16] or from intrinsic properties (low firing threshold, bursting propensity)[26]. CA3 is the main source of excitatory inputs to CA1 and is known to drive CA1 spatial representations in a majority of neurons, at least in familiar environments[48]. It was thus surprising to find that instant PFs were much less prevalent in CA3 than CA1 (Fig. 2). The simplest explanation is that CA1 instant PFs are not inherited from CA3 inputs during initial exploration. Indeed, not all CA1 place cells are necessarily driven by CA3[48] as CA1 receives other sources of spatially modulated inputs (entorhinal

cortex[4], CA2[10], non-imaged subareas of CA3[49], nucleus reuniens[50]), but the emergence dynamics of spatial representations in these areas are not currently known. Alternatively, CA1 instant PFs could be driven by the few CA3 neurons with instant PFs if those neurons have a high degree of divergence to CA1. Low dendritic inhibition in CA1 pyramidal cells upon initial exposure to novel environments could serve to amplify the influence of low numbers of CA3 inputs[15,25,51]. Delayed-onset CA3 neurons as well as neurons with unstable spatial modulation could also partially contribute to CA1 instant PFs since some of them are active on early trials.

After PF emergence, PF properties (position, width, shape) evolve with familiarization and their dynamics have been used as a proxy to study the synaptic plasticity mechanisms supporting spatial representations in the hippocampus[30,32,45]. Initial reports showed that, in a familiar environment, the population of CA1 PFs shifts backwards with experience, a phenomenon consistent with Hebbian rules[32,45] and dependent on NMDA-receptors[33]. Later electrophysiological studies compared lap-by-lap shifting dynamics in CA1 versus CA3 under different familiarity levels[29–31]. Despite discrepancies across studies attributable to different ways of defining novelty, our results are generally consistent with these studies, especially with Roth et al.[31] who used completely new distal and proximal cues. We observed significant backward shifting of the population of PFs on day 1 of a new virtual environment in CA1, and to a lesser extent CA3, with the

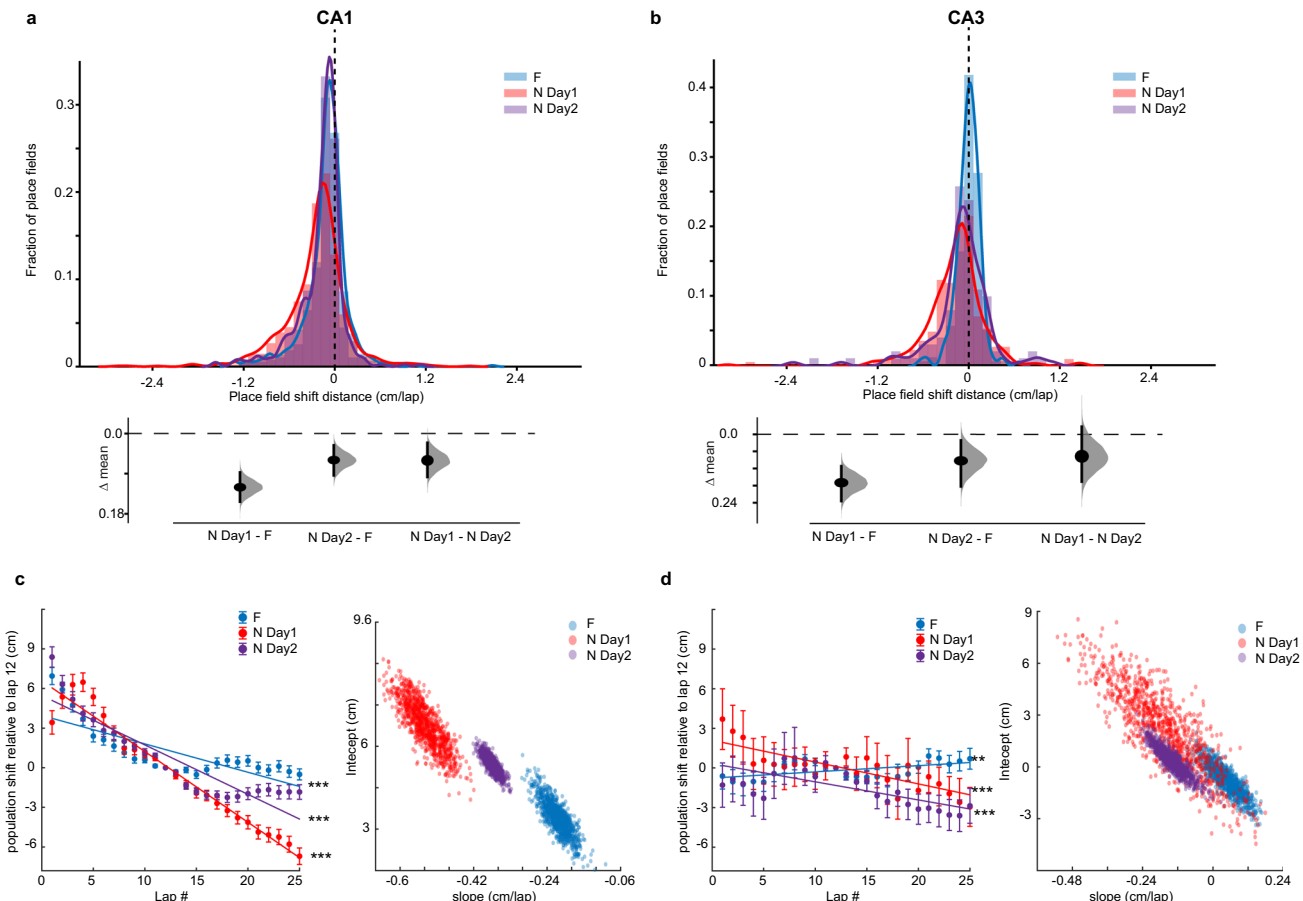

**Fig. 6 Population backward shifting slows down with familiarization across days. a** Top, histograms of $COM_{last\_lap}$–$COM_{onset\_lap}$ (session shift defined as in Fig. 3d) in Familiar (F), N day 1 and N day 2, for CA1. Bottom, bootstrapped mean differences (Δ) between the three conditions. **b** Same as a for CA3. **c** Population shift of PF COM. Left, mean ±SEM, over all PFs, of five-lap sliding average COM difference relative to lap 12 (see Methods), comparing CA1 population shifts in F (blue), N day 1 (red), and N day 2 (purple). Linear regression, F test, ***$P < 0.001$, F, $P = 3.1 \times 10^{-86}$, N Day 1, $P < 1 \times 10^{-100}$, N Day 2, $P < 1 \times 10^{-100}$. The resampling analysis uses a number of PFs equal to 80% of the number of PFs in the condition with the lowest number of PFs (1000 resamples). The lack of distribution overlap between conditions indicates dynamics are significantly different. **d** Same as c for CA3. Overlap in the slope distributions suggests that population dynamics are similarly flat (slopes close to 0) although CA3 PFs shifting become more homogeneous (smaller variance) with familiarization. Linear regression, F test, ***$P < 0.001$, F, $P = 0.005$, N Day 1, $P = 1.3 \times 10^{-4}$, N Day 2, $P = 1.1 \times 10^{-4}$.

shift slowing down with familiarization across days (Figs. 3–6). Consistent with most reports[31,32,45], we found that CA1 was still backward shifting in very familiar environments, but not CA3 (Fig. 6).

Population backward shifting was initially reported to coincide with increased negative skewness and enlargement of PFs[32,45]. However, in contrast to backward shifting, skewness and width dynamics are inconsistent across studies[30,31], suggesting heterogeneity of mechanisms leading to backward shifting[30,52]. Our data indicate that population backward shifting in CA1 in a novel environment is due to a combination of PF expansion, skewness changes, and pure translation of the PF (Supplementary Figs. 4, 8). These dynamics are dependent on familiarity, with changes in skewness closely matching population backward shifting. We did not detect clear dynamics in PF width or skewness in CA3, revealing a dissociation in mechanisms supporting population shifting in CA3 and CA1.

In addition to the population trend, we characterized shifting dynamics in individual PFs, revealing that half of PFs show significant linear shifts in CA1, mostly backward, compared to only a third in CA3 (Fig. 3c). PFs with large shifts tended to be less stable across days (Supplementary Fig. 10), suggesting that the same mechanisms may underly lap-to-lap and day-to-day stability. On

the other hand, PFs that significantly shifted on N day 1 but were stable across days tended, upon re-exposure on day 2, to reset their location toward their original position on day 1 (Fig. 5b, Supplementary Fig. 11). This reset of shifting shows that the plasticity underlying lap-by-lap shifting does not continue offline and is thus experience-dependent[32]. It also shows that lap-by-lap plasticity does not necessarily cause long-term changes in PFs. Combining these results, we hypothesize that synaptic plasticity underlying PF shifting may decay offline if too weak, promoting PF resetting the next day, but may have more lasting effects if stronger, promoting some forms of remapping across days.

Interestingly, we found that PFs did not shift backward in CA1 for the first few trials in the novel environment, they actually shifted forward on average (Supplementary Fig. 9). This fits with the idea that synapses from CA3 place cells could be necessary to initiate backward shifting in CA1, as only a very small number of CA3 PFs emerge on initial laps in the novel environment. As more CA3 PFs emerge those synapses may undergo asymmetric plasticity, such as spike-timing-dependent plasticity (STDP)[12] and behavioral time-scale plasticity (BTSP)[11,24], which could trigger backward shifting in CA1[11,30–33]. Indeed, both STDP and BTSP are known to occur at CA3–CA1 synapses of pyramidal neurons[11,12,24] but other synapses could also be involved.

A classic STDP rule fits well with the extent of backward shifting we observe across laps[32,52] as it can potentiate synapses activated up to 20 milliseconds prior to postsynaptic firing, which, based on the average running speed we observed, would maximally shift PFs backwards by ~0.5 cm/lap, approximately what we see in CA1 on N day 1. However, STDP is not always asymmetric at CA3–CA1 synapses, depending on induction protocols[12]. Moreover, the relevance of this type of time-dependent plasticity in vivo has been called into question as synaptic changes might just be too weak to yield an effect on single trials[53,54]. On the other hand, BTSP induces larger synaptic changes[24] and may thus be more likely to produce visible shifting effects over single trials. However, BTSP requires plateau potentials associated with burst firing, and those may not happen on most laps[25,55], which may be necessary to see continuous linear shifts as we observed here. BTSP can also lead to potentiation of synapses activated several seconds prior to burst firing, which could induce larger PF shifting than observed here. In addition, BTSP has only been found at CA3–CA1 synapses so far and may not occur in CA3 pyramidal neurons even though CA3 shows some backward shifting, albeit less than CA1. Lastly, there is evidence that cholinergic input from the medial septum could have a role in regulating backward shifting, at least in CA1[56]. It is therefore not yet clear which mechanisms are driving backward shifting in CA1 and CA3 place cells. The mechanisms supporting forward shifting in some neurons is also unknown. Our careful characterization of the shifting phenomenon in both CA1 and CA3 will constrain future computational models to clarify how hippocampal physiology supports plasticity of spatial representations.

The reinstatement of the same place code upon re-exposure to the same environment is generally thought to provide a neural substrate for remembering the environment[19–21,47]. Single unit recordings supported this idea by showing place cells remain stable over long periods[57]. However, recent evidence from calcium imaging of large ensembles of place cells has shown that many place cells in CA1 are unstable across exposures to the same environment[10,13,18,28,36–39]. Rather than a deterioration in the memory representation of the environment, such remapping may serve a function and has been proposed to encode distinct episodes that occur in the same environment separated by time or small changes in environment[10,37,39,41,58–62]. This remapping might therefore allow the animal to create independent representations of different episodes associated with the same environment and favor mnemonic discrimination of those episodes[63,64]. Our data support this idea as the average PF correlation in CA1 across days in a novel environment was 0.49 ± 0.02 (see methods) revealing considerable remapping across days (Fig. 4). This value is higher than what is reported in a recent study using methods similar to ours (~0.2)[18]. This discrepancy could come from differences in environment complexity which has been shown to influence PF stability and reliability[18,65], although no differences were observed between moderate and enriched visual environments, only impoverished environments showed reduced stability[18]. Differences in the experimental paradigm could also factor in: they switched repeatedly back and forth between novel and familiar environments within a single recording session, whereas our mice switched only once, defining only two different episodes (familiar and novel). Increased remapping due to repetitive switching, which defines several episodes, is consistent with the hypothesis of CA1 tracking distinct episodes.

A constantly changing place code in the hippocampus could nevertheless be problematic as information about the familiarity of the environment would be lost. CA3 has long been theorized to have a stable code to support memory recall through pattern completion via recurrent connections[7]. We show that CA3 place cells are indeed more stable across exposures to a novel environment than CA1 place cells. To our knowledge, the only other study to track CA3 place cells across days in a novel environment did not observe higher stability in CA3 (~0.1 spatial correlation in their study versus ~0.7 reported here in Fig. 4)[18]. In addition to the differences in task design stated above, this discrepancy could be explained by their inability to distinguish between CA3 and CA2 cells, as CA2 cells are known to have unstable place coding properties[10]. Another related study did find a similar effect as we report herein familiar environments[39] and concluded that CA3 provides a highly stable representation of space and context but little information about time, whereas CA1 is selectively required to integrate CA3 spatial/contextual and CA2 temporal information over hours and days. We add to this framework by showing that the reactivation of stable place cells in CA3 occurs rapidly (on the first traversal) upon re-exposure to the environment. This is consistent with the idea of CA3 recurrent connectivity supporting pattern completion through discrete attractors, with CA3 activity rapidly settling into the correct attractor basin upon re-exposure[5]. Transition dynamics from one attractor to the next likely depend on specific connectivity and excitability properties: identifying the characteristics of recurrent neural network models that would fit the fast transition we observed will be essential to understand the elusive mechanisms of hippocampal remapping.

Overall, the differences in emergence, shifting, and stability of PFs in CA1 and CA3 suggest distinct roles and mechanisms at play in these hippocampal subnetworks to support spatial memories. Instant CA1 PFs could mediate the ability of the hippocampus to rapidly represent episodes on a single trial, a feature of episodic memory[66]. The backward shifting of the PF population during ongoing experience could allow CA1 to gradually better predict future locations within an environment before physically arriving at those locations[11,67,68]. CA1, therefore, rapidly generates unique representations of the world that are then continuously updated by exploratory experience to predict the near future (where am I going?). In parallel, the CA3 gradually forms representations with stable trial-to-trial dynamics, thus encoding location in the present moment (where am I currently?). This function seemingly extends across time as relatively stable CA3 spatial representations are rapidly reinstated upon re-exposure to the same environment, possibly to support memory recall through pattern completion. Remapping of CA1 spatial representations across days may instead serve to separate events occurring in the same environment into distinct memory episodes[69]. This framework is depicted in a conceptual model in Supplementary Fig. 12.

## Methods

**Subjects**. All experimental and surgical procedures were in accordance with the University of Chicago Animal Care and Use Committee guidelines. For this study, 10–12 week old C57BL/6 J wildtype (WT) male mice (23–33 g) (four WT for CA1 population imaging, Jackson Lab 000664) and C57BL/6-Tg(Grik4-cre)G32-4Stl/J (seven for CA3 population imaging, Jackson Lab, 006474) were individually housed in a reverse 12 h light/dark cycle with an ambient temperature of ~20 °C and ~50% humidity. Male mice were used over female mice due to the size and weight of the headplates (9.1 mm × 31.7 mm, ~2 g), which were difficult to firmly attach to smaller female skulls. All training and experiments were conducted during the animal's dark cycle.

**Mouse surgery and virus injection**. Mice were anesthetized (~1–2% isoflurane) and injected with 0.5 mL of saline (intraperitoneal injection) and ~0.45 mL of meloxicam (1–2 mg/kg, subcutaneous injection). For CA1 population imaging, a small (~0.5–1.0 mm) craniotomy was made over the hippocampus CA1 (1.7 mm lateral, −2.3 mm caudal of Bregma). A genetically encoded calcium indicator, AAV1-CamKII-GCaMP6f (Addgene, #100834) was injected into CA1 (~75 nl) at a depth of 1.25 mm below the surface of the dura using a beveled glass micropipette. For CA3 population imaging, the craniotomy was made over the CA3 (2.0 mm lateral, −1.7 mm caudal of Bregma). A Custom made Cre-dependent AAV virus:

AAV1-CamKII-flex-GCaMP6f (made by Vigene) was injected (two injection sites at least 100 μm apart within the craniotomy, ~75 nl at each site) at a depth of 1.9 mm below the surface of the dura. After injection, the site was covered up using dental cement (Metabond, Parkell Coropration) and a metal head plate (Atlas Tool and Die Works). Water scheduling began the following day (0.8–1 ml per day and continued through all training and experiments). Around 7 days later, mice underwent another surgery to implant a hippocampal window as previously described[15]. Following implantation, the head plate was reattached with the addition of a head ring cemented on top of the head plate which was used to house the microscope objective and block out ambient light. For CA3 mice, because the cannula window was implanted at an angle (~15 degrees) relative to the horizontal plane, we bent the two ends of the head plate to match this angle so that the head plate and cannula were on the same plane. We could then change the angle of our microscope objective to be perpendicular to this plane. Post-surgery, mice were given 2–3 ml of water/day for 3 days to enhance recovery before returning to the reduced water schedule (0.8–1.0 ml/day).

**Behavior and virtual reality (VR) switching.** To navigate in the VR environment, animals ran on a treadmill surrounded by five LED screens[15,70]. VR environments (one training environment, which served as the familiar environment, F, and two novel environments: N1 and N2) were created using VIRMEn[71]. Each environment contained a three-meter-long linear track enriched with different distal and proximal 3D visual cues. In all, 4 μL water rewards were delivered at the end of the track in all environments. During training, which began at least 5 days after window implantation, mice were placed in F for 30–40 mins each day and learned to run and lick the water reward in F. After each lap traversal, mice were teleported back to the beginning of the track. Before teleportation, a short VR pause of 1.5 s was implemented to allow for water consumption and to help distinguish laps from one another rather than them being continuous. Once mice reached the criterion more than two laps per min that remained stable for 2–3 days (usually ~10–14 days after the start of training), imaging commenced.

**Two-photon imaging.** Imaging was done using a laser scanning two-photon microscope (Neurolabware). The microscope consisted of an 8 KHz resonant scanning module (Thorlabs), a 16×/0.8 NA/3 mm WD water immersion objective (MRP07220, Nikon). GCaMP6f was excited at 920 nm with a femtosecond-pulsed two-photon laser (Insight DS + Dual, Spectra-Physics) and the fluorescence was collected using a GaAsP PMT (H11706, Hamamatsu). The microscope is customized to tilt the objective, which we tilted to be perpendicular to the CA3 head plate angle but kept vertical for CA1 imaging. Stray light from the VR monitor was blocked from entering the objective lens by a dark rubber tube attached to the implanted head ring and the objective. Laser average power after the objective was ~60 mW for CA1 imaging and ~120 mW for CA3 to gain similar baseline fluorescence levels in the CA1 or CA3 FOV. Scanbox (Neurolabware) was used for microscope control and data acquisition. Time-series videos were acquired at around 11 Hz for each of the three imaging planes (using an electronic lens) to maximize the number of neurons imaged in each mouse. The PicoScope Oscilloscope (PICO4824, Pico Technology) collected the signal from the microscope to synchronize frame acquisition timing with behavior (see below).

**Imaging sessions.** Each mouse that reached the behavior criterion was carefully checked for expression under the two-photon microscope. Each mouse used in this study had healthy-looking GCaMP6f expression (resting fluorescence absent from the nucleus; fast transient kinetics; no signs of misshaped somas). On experimental day 1: fields of view (FOV) were chosen that maximized the number of neurons across three planes. Imaging and behavior recordings started right before mice entered the VR. Mice ran at least 20 laps in F, which took at least 10 min. After which the mice were instantaneously switched to a novel environment (N1). Mice then ran at least 35 laps in N1 and were recorded for at least 20 min and then placed back in their home cage. Experimental day 2: a similar procedure whereby mice were exposed to F first and then switched to a novel environment, but this novel environment (N2) was different from the first novel environment (N1). The FOVs were not necessarily the same as day 1. After imaging, more than one averaged FOV was saved to be the reference for day 3 imaging in order to align the planes and record from the same cells the following day. Experimental day 3: the same FOVs were carefully matched to the previous day FOVs. Once imaging started, mice were directly exposed to N2 and ran for at least 29 laps, and recorded for 20 min. Mice behavior including treadmill running speed, position, and licking was collected using the PicoScope Oscilloscope to synchronize with the imaging.

**Image processing and ROI selection.** Time-series movies for multiplane recordings were acquired using interleaved frames (1st, 4th, 7th… frames belong to plane 1; 2nd, 5th, 8th… frames belong to plane 2: 3rd, 6th, 9th… frames belong to plane 3). Each multiplane time series was then split into separate time-series movies. Same plane movies from Day 1 in F and N1 were concatenated into one movie, as were Day 2 single plane movies in F and N2, and Day 2 N2 and Day 3 N2 single plane movies (for across days analysis of the same cells). Movement artifacts are corrected by customized MATLAB scripts based on whole frame cross-correlation. For multiday imaging datasets (Day 2 N2 and Day 3 N2

concatenation), motion correction was applied before concatenation and then Fiji (ImageJ) was used to correct any rotational displacement between the two movies. The concatenated movies were then motion corrected again to assure the best alignment (Fig. 4).

Regions of interest (ROIs) were defined using customized MATLAB scripts from the Dombeck lab[15] (parameters: mu = 0.6, 150 principal components, 150 independent components, s.d. threshold = 2.5, s.d. smoothing width = 1, area limits = manually chose for each FOV). For each ROI, baseline-corrected ΔF/F traces across time, filtered for significant calcium transients were then generated as previously described[13,15,46]. In brief, slow time-scale changes in the fluorescence time series were removed by examining the distribution of fluorescence in a ±5 second interval around each sample time point and subtracting the 8% percentile value. The baseline and σ were calculated from the fluorescence time series that did not contain large transients. Fluorescence transients were then identified as events that started when fluorescence deviated 2σ from the corrected baseline, and ended when it returned to within 0.5σ of baseline. The baseline-subtracted neuron fluorescence traces were then subjected to analysis of the ratio of positive- to negative-going transients of various amplitudes and durations. We used this analysis to identify significant transients with <1% false-positive error rates and generated the significant transient-only traces that were used for all subsequent analysis.

**Calcium transient analysis.** After extracting significant calcium transients, we analyzed and compared some basic characteristics of these transients across CA1 and CA3. Transient peaks: the maximum value for each transient from each neuron. Transient duration: the duration of each transient calculated at half peak from each neuron. Transient frequency: the frequency of significant transients from each neuron.

**Behavior analysis.** First, immobile and backward moving periods were removed by identifying instantaneous velocity signals slower than 0.2 cm/s. Second, to calculate the mean lap velocity on each lap, we divided the track length (3 m) by the time taken to finish the lap. Third, to then calculate normalized mean lap velocity (Figs. 1b; 4b), we took the mean lap velocity on each lap and divided it by the mean velocity of the first three laps in F.

**Defining PFs.** Because mice ran continuously and consistently in all conditions, we included all laps and transients for PF identification. The 3 m track was divided into 50 bins (6 cm per bin). The mean ΔF/F was calculated as a function of virtual track position for 50 position bins for each lap, which formed a 50 by N laps matrix. Potential PFs were first identified as contiguous points of this matrix in which all of the points were >15% of the difference between the peak ΔF/F value (from all 50 bins) and the baseline value (mean of the lowest 12 out of 50 ΔF/F values). The potential PF had to satisfy the following criteria to be defined as a significant PF: 1. The field width must be >20 cm and <150 cm. 2. The field must have at least one value bigger than 0.1 ΔF/F. 3. The mean in field ΔF/F value must be greater than three times the mean out of field ΔF/F value. 4. Significant calcium transients must be present on at least 15 laps out of all the laps that the mouse traversed. Potential PF regions that met these criteria were then defined as PFs if their P value from boot strapping was <0.05, as described previously[46]. PFs from cells that have multiple PFs used the same criteria and were treated independently. Transients that occurred outside of the defined PF region were removed for analysis of each specific field. The resultant PFs were then used in all subsequent analyses unless specified.

**Histology and brain slices imaging.** We checked the CA3 expression of some of the Grik4-cre mice to ensure the GCaMP expression was restricted to CA3. Mice were anesthetized with isoflurane and perfused with ~10 ml phosphate-buffered saline (PBS) followed by ~20 ml 4% paraformaldehyde in PBS. The brains were removed and immersed in 30% sucrose solution overnight before being sectioned at 50 μm-thickness on a cryostat. The brain slices were then collected on glass slides and mounted with a mounting media with DAPI (SouthernBiotech DAPI-Fluoromount-G Clear Mounting Media, 010020). The whole-brain slices were imaged under ×10 with a Caliber I.D. RS-G4 Large Format Laser Scanning Confocal microscope from the Integrated Light Microscopy Core at the University of Chicago.

**Spatial correlation.** To measure PF spatial correlation across environments, we found place cells that had PFs in either environment and then calculated the Pearson's correlation coefficient between the mean activity along the track (in 50 bins) for all laps in two environments. To measure the PF correlation within environment, we divided the session up into two halves based on the total number of laps completed. We then calculated the mean activity along the track for each half and calculated Pearson's correlation coefficient. For cells with multiple PFs, only the first PF on the track was included. To measure PF spatial correlation across days, we found place cells that had PFs in both days and then calculated the Pearson's correlation coefficient between the mean activity along the track (in 50 bins) for the last 10 laps in N day 1 and the first 10 laps in N day 2.

**PF onset lap.** To determine PF onset lap (Figs. 2c–e, 4h), starting from lap 1 we searched lap-by-lap for a lap with a significant calcium transient present within the boundaries of the future PF calculated from all the laps in the session. Once the lap was found, we would then search for significant calcium transients on each of the next five laps. If three of the six laps had significant calcium transients within the PF boundaries, that would be considered the PF onset lap, if not, we move to the next lap and repeated the analysis. If we changed this criterion and instead used two out of six laps or four out of six laps to define PF onset lap, the differences in distributions we observed between CA1 and CA3 remained. To control for the different numbers of laps that the mouse ran in F and N, the comparison in Fig. 2d only included the first 25 laps in F and N. If we instead included the later laps in N, the result did not change.

**PF COM and spatial precision.** To calculate the spatial precision, we first calculated the somatic transient COM on each traversal along the linear track. We measured ΔF/F in each bin. We then used the following equation to calculate the COM for each traversal $n$ ($COM_n$):

$$COM_n = \frac{\sum_i DF_i \cdot x_i}{\sum_i DF_i}$$

Where $DF_i$ is the somatic ΔF/F in bin $i$ and $x_i$ is the distance of bin $i$ from the start of the track. We then calculated the weighted average COM ($COM_w$) from all traversals $n$ ($COM_n$ from each traversal was weighted by the peak transient ΔF/F on that traversal ($A_n$)):

$$COM_w = \frac{\sum_n A_n \cdot COM_n}{\sum_n A_n}$$

Spatial precision[13] (SP) was then calculated as follows (inverse of the COM standard deviation):

$$SP = \frac{1}{\sqrt{\frac{\sum_n A_n (COM_n - COM_w)^2}{\sum_n A_n}}}$$

**Out/in PF firing ratio.** This was computed as the ratio between the mean ΔF/F in bins outside the PF and the mean ΔF/F in bins within the PF.

**Position decoding analysis.** Based on a recent study[72], we chose to use the long short-term memory (LSTM) neural network model to test whether representations in CA1 were better at decoding position on the first lap of N than CA3 representations.

Considering the differences in the number of PFs, we measured in CA1 and CA3 (more PFs in CA1 than CA3), as well as the amount of data required to build a useful LSTM model, we grouped data from all CA1 and CA3 mice that ran >45 laps in N2, and matched the number of cells used from CA1 and CA3 to build the models.

To match the data from different mice and use the decoders to decode position on a lap-by-lap basis, we first changed the time series-based data to position-based data: the 3 m track was divided into 100 bins (3 cm per bin). The mean ΔF/F was calculated as a function of virtual track position for 100 position bins for each lap. By doing this, data within and across mice became the same length by lap. CA1 data were then grouped into one data set and CA3 data grouped into another data set. The decoding data were restricted to periods when the animals were running.

To test decoding ability on the first lap in N, we first tested different parameters (LSTM model network structure and the number of place cells used to build the models) to make sure that the two decoders had similar decoding ability in the later laps (validation set) (Supplementary Fig. 3). We chose a one-layer LSTM decoder with 1024 units to decode the animals' position from the input of 200 place cells. When building a model, 200 place cells were randomly chosen from the entire CA1 or CA3 place cell population, the data from the 6th to 35th laps were used to train the decoder to decode the animals' position on the track which had been divided to 50 bins. The 36th to 40th lap data were used as the validation set. The decoders were then used to decode the animals' position on the first lap based on the place cell activity on this lap. We repeated this procedure 20 times for CA1 and CA3 and compared the predicting error on the first lap between the regions. The predicting error was calculated as the mean of the absolute difference between the prediction position and the animals' real position on the first lap.

We also built a naive Bayes decoder with the same data, though the decoding ability is not as good as the LSTM decoder for the validation set, we got the same result as the LSTM decoder for the first lap position decoding (that is the CA1 is better at decoding position on the first lap compared to CA3) (Supplementary Fig. 3).

**PF shifting.** To calculate population PF shifting (Figs. 3e, 5d, 6c–d), we first calculate the COM for all PFs on each lap with a sliding window of five laps (the $COM_w$ of the current and four next laps). The five-lap sliding average was done for smoothing purposes but the same trends were observed without smoothing. For each PF, lap-wise shift was computed as the difference between lap 12 and the current lap. We could then calculate the average shifting over the population of PFs on each lap. Only PFs with PF activity on lap 12 and a PF onset lap <20 were included. Also, owing to the onset of PFs on different laps, the number of samples that contribute to the mean on each lap is different.

Note that we also used the five-lap COM weighted average method in Fig. 3d.

**PF skewness.** PF skewness is calculated as the third statistical moment of the PF. For lap $n$:

$$Skewness_n = \sum_i \frac{DF_i}{\sum_i DF_i} \cdot \frac{(x_i - COM_n)^3}{\sigma^3}$$

where $\sigma$ is the PF's "standard deviation":

$$\sigma = \sqrt{\sum_i \frac{DF_i}{\sum_i DF_i} \cdot (x_i - COM_n)^2}$$

To calculate population place skewness trend, we first calculate the skewness for all PFs on each lap. Then we took each PF and aligned them together by the actual laps.

**PF width.** PF width on each lap is calculated as the difference between the first and last bin with in-field activity. We then normalized the lap-wise width to the mean PF width for each PF.

Notice, for COM shift and reset, PF skewness and width analyses, most PFs near start or end of the track were excluded if they were clipped, using the following criterion: if the distance between one track edge and PF COM was at least one bin shorter than the other half of the PF, the PF was excluded. For cells with multiple PFs, only the first PF on the track was included.

**Statistics and reproducibility.** Error in the text and figures are presented as mean ±SEM, unless stated otherwise.

We used either an estimation approach or null-hypothesis testing to compare data (described in figure legends). To generate Gardner-Altman estimation plots, which highlight the effect size, we used the Data Analysis with Bootstrapped-coupled ESTimation (DABEST) package[73] (available on GitHub: https://github.com/ACCLAB/DABEST-python). To assess the uncertainty of the effect size, the mean difference between two distributions and its 95% confidence interval were bootstrapped. For null-hypothesis testing, Wilcoxon rank-sum test, Wilcoxon signed-rank test or Kolmogorov–Smirnov two-sample test were applied. $P < 0.05$ was chosen to indicate statistical significance and $P$ values in figures are indicated as follows: *$P < 0.05$, **$P < 0.01$, ***$P < 0.001$, N.S. not significant. For data tested with the estimation approach, we also used the null-hypothesis testing to confirm any differences.

For linear regressions of the population slopes (e.g., Figure 3e left), the $p$ value for the $t$ statistics to test whether the slope was significantly positive or negative were reported following the same approach reported above. To compare the population dynamics of different conditions, we performed exact testing based on Monte-Carlo resampling (1000 resamples with sample size matching the lower sample size condition) as detailed in legends.

To assess the shifting dynamics in single PFs, we performed linear regression on the lap-wise $COM_n$ relative to onset lap (Fig. 3c), and significance was assessed with an F test.

Although Fig. 1c, d only shows representative images, we reproduced similar images multiple times across $n > 4$ mice. Figure 4a also only shows a representative image of a field of view (FOV) aligned across days. However, for all across days imaging, FOVs from all mice included in the subsequent analysis were aligned similar to Fig. 4a ($n = 3$ for CA1; $n = 6$ for CA3).

**Data and software.** Data processing and analysis was performed using custom written scripts in MATLAB (2018a) or Python (3.7).

**Reporting summary.** Further information on research design is available in the Nature Research Reporting Summary linked to this article.

## Data availability
Raw imaging data are extremely large and not feasible for upload to an online repository but is available upon request. Processed data are available on GitHub (https://github.com/Candong/Distinct_CA1_CA3.git)[74]. Source data are provided with this paper.

## Code availability
Scripts used for data analysis and processed data are available on GitHub (https://github.com/Candong/Distinct_CA1_CA3.git)[74].

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

## Acknowledgements
We thank D. Dombeck for comments on the manuscript, C. Cherian for helpful manuscript edits, and members of the Sheffield laboratory for manuscript comments and useful discussions. This work was supported by: The Whitehall Foundation, The Searle Scholars Program, The Sloan Foundation, The University of Chicago Grossman Institute for Neuroscience start-up funds, and the NIH (1DP2NS111657-01).

## Author contributions

Conceptualization: M.S., data curation: C.D., formal analysis: C.D., A.M., M.S., funding acquisition: M.S., investigation: C.D., methodology: M.S., C.D., A.M., project administration: C.D., software: C.D. supervision: M.S., A.M., validation: A.M., M.S., visualization: C.D., A.M., M.S., writing—original draft: M.S., C.D., writing—review & editing: A.M., M.S., C.D.

## Competing interests

The authors declare no competing interests.
