## [Peer Review File · Nature Communications]

Reviewers' Comments:

Reviewer #2:

Remarks to the Author:

In this study, trial-by-trial stability of place cell activity was compared between CA1 and CA3 in mice navigating through virtual reality on subsequent days. Particularly the emergence of place field activity upon exposure of mice to a novel environment was examined on two subsequent days. The main finding is that CA1 place cells form early and shift backwards with experience whereas in CA3 place fields emerge gradually but remain more stable across two days. The authors state that trial-by-trial dynamics across days during first moments of novelty experience on subsequent days has not been systematically examined in CA1 and CA3' (line 20-25). This is not correct; indeed several studies examined the emergence of place fields on several subsequent days (Rubin et al., *elife* 2015; Ziv et al., *Nature Neurosci* 2012; Hainmueller and Bartos, *Nature* 2018). Thus, this 'novelty' aspect requires substantial revision.

Main criticism:

1. Remapping of CA1 and CA3 place cells has been previously reported. Please refer to relevant publications (line 37, 38).
2. The place field size in CA3 as well as their trial-by-trial stability is dependent on the CA3 subarea (CA3a-c). It is therefore important to state in which CA3 sub-region principal cells have been imaged. The emergence of new place cells might also be dependent on the sub-field. Please comment on the place field size. New place fields seem to be larger than stable place fields (Fig. 3f, bottom).
3. Please provide information on the Grik4-cre line. What is the percentage of cells expression cre? Can we exclude some bias of labelling a subpopulation of CA3 principal cells?
4. Place fields in both CA1 and CA3 shifted on a lap-by-lap basis. The shift was more pronounced in CA1 than CA3. Although this is an interesting observation, it should be double-checked, whether this shift is real or generated by slight changes in the rotation of the Styrofoam ball caused by the running of mice. The synaptic interaction between CA3 and CA1 could in this case also support, as stated by the authors, the higher steepness in the observed shift in CA1 compared to CA3. The hypothesis that STDP might support backward shifts in CA1 place fields is interesting but requires validation. The arguments for plasticity underlying the observed emergence of place fields and their shift remains weak without further experimental validation.
5. What is the mean place field correlation of place fields between 1st and 2nd day of N exposure and how does it compare to published data? In general, rigorous comparisons of the findings with published work is required.

Reviewer #3:

Remarks to the Author:

In this study, Dong and Sheffield used two-photon imaging in head-fixed mice to examine the time course of development of CA1 and CA3 place-fields in virtual environments. While place-field formation in CA1 has been previously described, the current study benefits from an elegant experimental design in which place-field formation for both hippocampal regions are examined in parallel, using the same virtual environments to compare both regions on a familiar track and on consecutive runs on a novel track. The authors found that upon exposure to a new context, cells in CA1 formed place fields more quickly than cells in CA3, but that place fields in CA3 were more stable across and between sessions. The authors also reported that CA1 (more so than CA3) place fields shifted backwards over the course of a session, similar to some previous reports. Overall, this is nice work that will be impactful in furthering understanding of how these connected subregions coordinate to produce the hippocampal spatial code. The article is well written and carefully edited and succinctly describes the experiments and results. My comments mostly concern interpretation of the findings.

The authors report that CA1 forms place fields more rapidly than CA3 when an animal is exposed to a new context. Based on this observation, the authors argue that CA1 cells must be more driven by

direct entorhinal input rather than via CA3. However, it remains plausible to me that the few CA3 fields that do form instantaneously can be driving the fast-forming CA1 place-fields. Indeed, it's unlikely that the spatial representation in CA1 directly follows the representations in entorhinal cortex either, so if the same analysis was applied there, wouldn't the conclusion be reversed? Furthermore, it's not clear where the reported instabilities in the CA1 place-fields emerges from, since presumably the entorhinal cortical inputs are not changing similarly over these same periods. And if it is entorhinal cortex driving these fields, then why do the authors attribute the place-field shifts to plasticity at a different synapse, rather than these same ones? Finally, at least one recent study (Davoudi and Foster, 2019) suggests that CA3 drives CA1 place-fields. These authors should address these points in the discussion and provide a more detailed and less simplistic rationale for their interpretations of their findings.

The shifting of place-fields in novel environments in CA1 but not CA3 was another important finding in this study. The authors cite the important work of Mehta et al for this. However, there are some important differences between the current findings and those by Mehta et al that should be addressed. First, Mehta et al report a backward expansion of place-fields, not simply a shift. Thus the sizes of place-fields should also be increasing across trials. This doesn't appear to be the case here, which should be discussed. Second, Mehta et al only report this effect for place-fields on familiar tracks, not novel ones. As far as I am aware the effect has not been previously reported for place-fields on a novel track. In particular, the authors seem to indicate that the effect depends on novelty, though it appears to remain significant in a familiar context as well. A third important discrepancy is that in Mehta et al, the place-field expansion was reset at the onset of each recording day, and the place-fields began to shift and expand again over the course of the session, resetting again by the next day's session, presumably because of sleep. This would seem to contradict the findings here that the shifting place-fields remain in the same position from the end of one session to the start of the next (Fig 3h). I would recommend moving Extended Data Fig 6 to a main figure, adding more direct statistical examination of the curves in that figure, and providing a detailed discussion of these discrepancies between the findings here and the previous published studies.

Response to Peer-Review on manuscript NCOMMS-20-29326-T

We are writing to discuss our manuscript entitled "**Distinct place cell dynamics in CA1 and CA3 encode experience in new environments**". We would like to thank the editor and reviewers for the helpful feedback they provided us. Note that the initial submission was initially tailored as a Brief Communication for Nat. Neuroscience. We made major revisions to the manuscript: the main text has been completely rewritten and expanded to fit Nature Communications and we added several additional analyses in the main figures (reorganized compared to initial submission) as well as supplementary figures. Our responses to reviews are in blue. Line # in our response corresponds to the revised version of the manuscript. References cited here are listed at the end of the present document and do not correspond to the reference numbers in the manuscript.

Reviewer 1: In this study, trial-by-trial stability of place cell activity was compared between CA1 and CA3 in mice navigating through virtual reality on subsequent days. Particularly the emergence of place field activity upon exposure of mice to a novel environment was examined on two subsequent days. The main finding is that CA1 place cells form early and shift backwards with experience whereas in CA3 place fields emerge gradually but remain more stable across two days. The authors state that trial-by-trial dynamics across days during first moments of novelty experience on subsequent days has not been systematically examined in CA1 and CA3' (line 20-25). This is not correct; indeed several studies examined the emergence of place fields on several subsequent days (Rubin et al., *elife* 2015; Ziv et al., *Nature Neurosci* 2012; Hainmueller and Bartos, *Nature* 2018). Thus, this 'novelty' aspect requires substantial revision.

We would like to thank the reviewer for pointing out these important papers. We have now completely re-written and expanded the introduction and the discussion to include a much more detailed review of the literature. We think the new version now better highlights the specific components of our study that are novel in the context of the most relevant work in the area: we track large numbers of CA1 and CA3 neurons (using a transgenic line that allows to selectively target CA3 and distinguish from CA2, in contrast to Hainmueller and Bartos 2018¹) in novel environments across days (in contrast to most work that used already familiar environments, such as Ziv et al. 2012²) and we analyze the lap-by-lap dynamics of spatial representations emerging in novel environments, which has been done only in a few electrophysiological studies (see Roth et al. 2012³) which did not thoroughly investigate PF emergence dynamics and were not able to track the same neurons across days. Solving this technical challenge allowed us to carry out new analyses of single PF dynamics (Figs. 3c, 5a-c, Supplementary Figs. 5-7, 9-11) and to better understand the processes underlying the retrieval or remapping of spatial representations across exposures to the same environment.

1. Remapping of CA1 and CA3 place cells has been previously reported. Please refer to relevant publications (line 37, 38).

We have now edited this sentence to state that remapping has been previously reported and added a number of important references relevant to global remapping in CA1 and CA3. This is the new sentence and the papers referenced (Line 112-114):

“As has previously been reported, both CA1 and CA3 place cells globally remapped upon exposure to N (Fig. 2a, b) and displayed altered PF properties compared to F (Supplementary Fig. 2)^{1,4-10}.”

2. The place field size in CA3 as well as their trial-by-trial stability is dependent on the CA3 subarea (CA3a-c). It is therefore important to state in which CA3 sub-region principal cells have been imaged. The emergence of new place cells might also be dependent on the sub-field.

This is an important point. We are exclusively imaging from CA3a as our current imaging setup only allows access to this part of CA3. We have now made this clear in the first sentence of the results section (Line 98).

Please comment on the place field size. New place fields seem to be larger than stable place fields (Fig. 3f, bottom).

The original Fig. 3f did indeed show an example of a new place field on day 2 of N with a larger width than a stable place field that initially formed on day 1. Based on this reviewer’s observation we went back and measured the widths of all our place fields in the novel and familiar environment in CA1 and CA3 (Supplementary Fig. 2e). Indeed, it turns out to be true that place field widths are significantly larger in N than F. We added a comment about this difference to the results section (Line 114-115) and thank the reviewer for pointing this out.

3. Please provide information on the Grik4-cre line. What is the percentage of cells expression cre? Can we exclude some bias of labelling a subpopulation of CA3 principal cells?

The original Tonegawa article claims 100% of CA3 pyramidal cells are labeled with Cre in the Grik4 line. This is from their paper: “At 8 weeks of age, recombination had occurred in nearly 100% of pyramidal cells in area CA3 (Fig. 1C [their paper])”. We used 3-month-old mice in all our experiments. We therefore added this sentence to the start of the results section and referenced the original paper (Line 99-102):

“The Grik4-cre line¹¹ was used to restrict expression to CA3 pyramidal neurons (Fig. 1c). Importantly, these mice show recombination had occurred in nearly 100% of pyramidal cells in CA3 which means our recordings were not biased to a sub-population of CA3 pyramidal cells¹¹.”

4. Place fields in both CA1 and CA3 shifted on a lap-by-lap basis. The shift was more pronounced in CA1 than CA3. Although this is an interesting observation, it should be double-checked, whether this shift is real or generated by slight changes in the rotation of the Styrofoam ball caused by the running of mice.

We apologize to the reviewer for not making this point crystal clear in the main body of the paper, but our mice are not running on a Styrofoam ball which could indeed have rotated in the yaw axis and change the animal’s trajectory from trial to trial. We instead used a treadmill that only allowed rotations in the forward/backward direction making rotations in the yaw axis impossible. Because mouse behavior in this setup is very consistent from trial-to-trial (the only change in behavior we see is on the very first trial in N, see Fig. 1b), behavioral changes cannot

explain the trial-to-trial shifting dynamics in PFs. Even so we did analyze whether trial velocity was connected to the amount of PF shifting on the subsequent lap (see Supplementary Fig 6). We report in the legend that “This analysis does not reveal an obvious relationship between velocity and shifting and clearly shows that large lap-to-lap shifts are not due to higher velocities”.

To make the point clearer that mice are running on a 1D treadmill we added a sentence to the summary of results paragraph in the introduction (Line 80-83): “In this work we use 2-photon Ca^{2+} imaging to longitudinally record from large populations of CA3 and CA1 pyramidal neurons in head-fixed mice running unidirectionally on a treadmill to repeatedly traverse visually enriched virtual linear environments with consistent behavior.” To further make this clearer we also edited a sentence to the first paragraph of the results section (Line 108-111): “Because mice were restricted to running in 1 dimension on a custom-built treadmill this paradigm led to many repeated traversals in both environments with matched behavior, allowing lap-by-lap PF dynamics to be measured systematically and compared across F and N environments without confounds caused by changes in behavior.”

The synaptic interaction between CA3 and CA1 could in this case also support, as stated by the authors, the higher steepness in the observed shift in CA1 compared to CA3. The hypothesis that STDP might support backward shifts in CA1 place fields is interesting but requires validation. The arguments for plasticity underlying the observed emergence of place fields and their shift remains weak without further experimental validation.

We believe the reviewer has brought up a very important criticism of the paper that we have taken very seriously. As we now state in the discussion (Line 271-3): “PF properties (position, width, shape) [...] dynamics have been used as a proxy to study the synaptic plasticity mechanisms supporting spatial representations in the hippocampus¹²⁻¹⁴”. However, the reviewer is correct that we have not directly tested the hypothesis that STDP (or another plasticity mechanism such as BTSP) is causing the PF shifts we see, even though our data are consistent with this mechanism. We have removed this claim from the results section and now just discuss it as a potential mechanism along with BTSP in the discussion section, with a more extensive discussion of past experimental findings¹⁵ and computational modeling^{14,16} investigating the role of synaptic plasticity in PF shifting (Line 317-336). All reference to PF emergence being caused by synaptic plasticity have been moved to the discussion section as well. We also reduced the strength of our claim that STDP is the most likely mechanism to cause shifts and instead discuss it as one potential mechanism among others (BTSP, septal cholinergic signals).

5. What is the mean place field correlation of place fields between 1st and 2nd day of N exposure and how does it compare to published data? In general, rigorous comparisons of the findings with published work is required.

We have substantially expanded our discussion to include not only a comparison of PF stability with other relevant studies, but a thorough contextualization of all our results. Specifically, addressing the first part of the reviewer’s concern, we have added the mean \pm SEM values for PF correlations across days in CA1 (0.49 ± 0.02) and CA3 (0.70 ± 0.04) (distribution reported in Fig 4d) and compared these values with other reports in the discussion section (Line 348-59 and Line 365-73).

Reviewer #2: In this study, Dong and Sheffield used two-photon imaging in head-fixed mice to examine the time course of development of CA1 and CA3 place-fields in virtual environments. While place-field formation in CA1 has been previously described, the current study benefits from an elegant experimental design in which place-field formation for both hippocampal regions are examined in parallel, using the same virtual environments to compare both regions on a familiar track and on consecutive runs on a novel track. The authors found that upon exposure to a new context, cells in CA1 formed place fields more quickly than cells in CA3, but that place fields in CA3 were more stable across and between sessions. The authors also reported that CA1 (more so than CA3) place fields shifted backwards over the course of a session, similar to some previous reports. Overall, this is nice work that will be impactful in furthering understanding of how these connected subregions coordinate to produce the hippocampal spatial code. The article is well written and carefully edited and succinctly describes the experiments and results. My comments mostly concern interpretation of the findings.

The authors report that CA1 forms place fields more rapidly than CA3 when an animal is exposed to a new context. Based on this observation, the authors argue that CA1 cells must be more driven by direct entorhinal input rather than via CA3. However, it remains plausible to me that the few CA3 fields that do form instantaneously can be driving the fast-forming CA1 place-fields. Indeed, it's unlikely that the spatial representation in CA1 directly follows the representations in entorhinal cortex either, so if the same analysis was applied there, wouldn't the conclusion be reversed? Furthermore, it's not clear where the reported instabilities in the CA1 place-fields emerges from, since presumably the entorhinal cortical inputs are not changing similarly over these same periods. And if it is entorhinal cortex driving these fields, then why do the authors attribute the place-field shifts to plasticity at a different synapse, rather than these same ones? Finally, at least one recent study (Davoudi and Foster, 2019) suggests that CA3 drives CA1 place-fields. These authors should address these points in the discussion and provide a more detailed and less simplistic rationale for their interpretations of their findings.

Our discussion is completely re-written and we do not make strong claims about which synapses are involved anymore. We now discuss all points and studies brought up by the reviewer more thoroughly and cautiously in the second paragraph of the discussion (Line 254-269):

“Rapid PF emergence in a novel environment in CA1 is thought to rely on a combination of strong, spatially tuned excitatory inputs that do not need to be potentiated¹⁷ and high neuronal excitability, either from novelty-induced disinhibition¹⁸ or from intrinsic properties (low firing threshold, bursting propensity)¹⁹. CA3 is the main source of excitatory inputs to CA1 and is known to drive CA1 spatial representations in a majority of neurons, at least in familiar environments²⁰. It was thus surprising to find that instant PFs were much less prevalent in CA3 than CA1 (Fig 2). The simplest explanation is that CA1 instant PFs are not inherited from CA3 inputs during initial exploration. Indeed, not all CA1 place cells are necessarily driven by CA3²⁰ as CA1 receives other sources of spatially modulated inputs (entorhinal cortex⁶, CA2²¹, non-imaged subareas of CA3²², nucleus reuniens²³), but the emergence dynamics of spatial representations in these areas are not currently known. Alternatively, CA1 instant PFs could be driven by the few CA3 neurons with instant PFs if those neurons have a high degree of divergence to CA1. Low dendritic inhibition in CA1 pyramidal cells upon initial exposure to

novel environments could serve to amplify the low number of CA3 inputs^{7,24,25}. Delayed onset CA3 neurons as well as neurons with unstable spatial modulation could also partially contribute to CA1 instant PFs since some of them are active on early trials.”

The shifting of place-fields in novel environments in CA1 but not CA3 was another important finding in this study. The authors cite the important work of Mehta et al for this. However, there are some important differences between the current findings and those by Mehta et al that should be addressed. First, Mehta et al report a backward expansion of place-fields, not simply a shift. Thus the sizes of place-fields should also be increasing across trials. This doesn't appear to be the case here, which should be discussed. Second, Mehta et al only report this effect for place-fields on familiar tracks, not novel ones. As far as I am aware the effect has not been previously reported for place-fields on a novel track. In particular, the authors seem to indicate that the effect depends on novelty, though it appears to remain significant in a familiar context as well.

We are glad that the reviewer motivated us to analyze our data in this way as it has led to interesting results that have now been added to the paper (Supplementary Fig. 8, Line 142-152 and 200-217). We provide a thorough discussion of these findings in the context of the few past electrophysiological studies that investigated the matter (Line 271-93):

“After PF emergence, PF properties (position, width, shape) evolve with familiarization and their dynamics have been used as a proxy to study the synaptic plasticity mechanisms supporting spatial representations in the hippocampus¹²⁻¹⁴. Initial reports showed that, in a familiar environment, the population of CA1 PFs shifts backwards with experience, a phenomenon consistent with Hebbian rules^{12,14} and dependent on NMDA-receptors¹⁵. Later electrophysiological studies compared lap-by-lap shifting dynamics in CA1 versus CA3 under different familiarity levels^{3,13,26}. Despite discrepancies across studies attributable to different ways of defining novelty, our results are generally consistent with these studies, especially with Roth et. al. (2012)³ who used completely new distal and proximal cues. We observed significant backward shifting of the population of PFs on day 1 of a new virtual environment in CA1, and to a lesser extent CA3, with the shift slowing down with familiarization across days (Figs. 3, 5, 6). Consistent with most reports^{3,12,14}, we found that CA1 was still backward shifting in very familiar environments, but not CA3 (Fig. 6).

Population backward shifting was initially reported to coincide with increased negative skewness and enlargement of PFs^{12,14}. However, in contrast to backward shifting, skewness and width dynamics are inconsistent across studies^{3,13}, suggesting a heterogeneity of mechanisms leading to backward shifting^{13,16}. Our data indicates that population backward shifting in CA1 in a novel environment is due to a combination of PF expansion, skewness changes and pure translation of the PF (Supplementary Figs. 4, 8). These dynamics were dependent on familiarity, with changes in skewness closely matching population backward shifting. We did not detect clear dynamics in PF width or skewness in CA3, revealing a dissociation in mechanisms supporting population shifting between CA3 and CA1.”

A third important discrepancy is that in Mehta et al, the place-field expansion was reset at the onset of each recording day, and the place-fields began to shift and expand again over the course of the session, resetting again by the next day's session, presumably because of sleep. This

would seem to contradict the findings here that the shifting place-fields remain in the same position from the end of one session to the start of the next (Fig 3h).

This is an interesting point. We are not aware of past studies investigating PF resetting across days (as far as we know, it is very difficult to track the same single units across days with tetrode recordings), but Mehta et al.¹⁴ did compare PFs skewness on the last lap of a familiar track and the first lap of a different familiar track, recorded right after the session in the first track. In our initial submission we had performed a similar analysis comparing PF location at the end of the session on day 1 and the beginning of the session on day 2 in a novel environment for all PFs stable across days (now in Fig 5a) and, unlike Mehta with skewness in familiar environments, we found no significant difference on average. Note that we focused on PF center of mass location rather than skewness like Mehta because skewness dynamics are inconsistent across studies and measuring the shift of the center of mass captures the full dynamics of changes (combination of changes in width, skewness and translation)^{3,13,16}. In the current version of our manuscript, we expanded on our initial findings by focusing on PFs with significant shifting on day 1 (Fig 5b) and found that, in this case, many PFs with large shifts on day 1 reset in the direction opposite of their shift on day 1, consistent with Mehta et al. (2000). Like for skewness across familiar environments¹⁴, PFs do not reset necessarily exactly to their initial position on day 1 (Supplementary Fig. 11). Those results are reported on lines 185-194 and discussed in the context of past research on lines 295-306.

I would recommend moving Extended Data Fig 6 to a main figure, adding more direct statistical examination of the curves in that figure, and providing a detailed discussion of these discrepancies between the findings here and the previous published studies.

We followed the reviewer's advice and moved this figure to the main text. We also expanded the analysis to now include distribution plots to show the entire population of place fields in CA1 and CA3 in each condition (Nday1, Nday2, F) with formal comparison of the means using bootstrapped estimation plots (Fig. 6a, b). To further analyze population dynamics and formally compare them across subfields and conditions throughout the article, we used Monte-Carlo resampling and exact testing^{27,28} (a non-parametric hypothesis testing approach akin to the bootstrap) (see Fig. 3e legend), which controls that different dynamics are not due to differences in sample size. The variance of the resampled distributions is also a good indicator of the homogeneity of dynamics across the population, which we comment throughout the paper. In Fig 6, this analysis clearly shows how the extent of backward shifting is dependent on familiarity with the environment in CA1 and much less so in CA3 which shows a lot of overlap in each condition. Based on an above comment from this reviewer, we added a similar analysis on the development of negative skewness and width (Supplementary Fig. 8). Finally, to get a better understanding of the variability and processes underlying population shifting, we added a thorough analysis of the shifting dynamics in individual place fields (Fig. 3c) and how it relates to different behavioral and neuronal variables in different conditions and hippocampal subfields (Supplement Fig. 5-7, 9-11).

References

- 1 Hainmueller, T. & Bartos, M. Parallel emergence of stable and dynamic memory engrams in the hippocampus. *Nature* **558**, 292-296, doi:10.1038/s41586-018-0191-2 (2018).
- 2 Ziv, Y. *et al.* Long-term dynamics of CA1 hippocampal place codes. *Nat Neurosci* **16**, 264-266, doi:10.1038/nn.3329 (2013).
- 3 Roth, E. D., Yu, X., Rao, G. & Knierim, J. J. Functional differences in the backward shifts of CA1 and CA3 place fields in novel and familiar environments. *PLoS One* **7**, e36035, doi:10.1371/journal.pone.0036035 (2012).
- 4 Colgin, L. L., Moser, E. I. & Moser, M. B. Understanding memory through hippocampal remapping. *Trends Neurosci* **31**, 469-477, doi:10.1016/j.tins.2008.06.008 (2008).
- 5 Jeffery, K. J. Place cells, grid cells, attractors, and remapping. *Neural Plast* **2011**, 182602, doi:10.1155/2011/182602 (2011).
- 6 Latuske, P., Kornienko, O., Kohler, L. & Allen, K. Hippocampal Remapping and Its Entorhinal Origin. *Front Behav Neurosci* **11**, 253, doi:10.3389/fnbeh.2017.00253 (2017).
- 7 Sheffield, M. E. J., Adoff, M. D. & Dombeck, D. A. Increased Prevalence of Calcium Transients across the Dendritic Arbor during Place Field Formation. *Neuron* **96**, 490-504 e495, doi:10.1016/j.neuron.2017.09.029 (2017).
- 8 Muller, R. U. & Kubie, J. L. The effects of changes in the environment on the spatial firing of hippocampal complex-spike cells. *J Neurosci* **7**, 1951-1968 (1987).
- 9 Bostock, E., Muller, R. U. & Kubie, J. L. Experience-dependent modifications of hippocampal place cell firing. *Hippocampus* **1**, 193-205, doi:10.1002/hipo.450010207 (1991).
- 10 Fyhn, M., Hafting, T., Treves, A., Moser, M. B. & Moser, E. I. Hippocampal remapping and grid realignment in entorhinal cortex. *Nature* **446**, 190-194, doi:10.1038/nature05601 (2007).
- 11 Nakazawa, K. *et al.* Requirement for hippocampal CA3 NMDA receptors in associative memory recall. *Science* **297**, 211-218, doi:10.1126/science.1071795 (2002).
- 12 Mehta, M. R., Barnes, C. A. & McNaughton, B. L. Experience-dependent, asymmetric expansion of hippocampal place fields. *Proc Natl Acad Sci U S A* **94**, 8918-8921, doi:10.1073/pnas.94.16.8918 (1997).
- 13 Lee, I., Rao, G. & Knierim, J. J. A double dissociation between hippocampal subfields: differential time course of CA3 and CA1 place cells for processing changed environments. *Neuron* **42**, 803-815, doi:10.1016/j.neuron.2004.05.010 (2004).
- 14 Mehta, M. R., Quirk, M. C. & Wilson, M. A. Experience-dependent asymmetric shape of hippocampal receptive fields. *Neuron* **25**, 707-715, doi:10.1016/s0896-6273(00)81072-7 (2000).
- 15 Ekstrom, A. D., Meltzer, J., McNaughton, B. L. & Barnes, C. A. NMDA receptor antagonism blocks experience-dependent expansion of hippocampal "place fields". *Neuron* **31**, 631-638, doi:10.1016/s0896-6273(01)00401-9 (2001).
- 16 Yu, X., Knierim, J. J., Lee, I. & Shouval, H. Z. Simulating place field dynamics using spike timing-dependent plasticity. *Neurocomputing* **69**, 1253-1259 (2006).

- 17 Sheffield, M. E. & Dombeck, D. A. Dendritic mechanisms of hippocampal place field formation. *Curr Opin Neurobiol* **54**, 1-11, doi:10.1016/j.conb.2018.07.004 (2019).
- 18 Pedrosa, V. & Clopath, C. The interplay between somatic and dendritic inhibition promotes the emergence and stabilization of place fields. *PLoS Comput Biol* **16**, e1007955, doi:10.1371/journal.pcbi.1007955 (2020).
- 19 Epsztein, J., Brecht, M. & Lee, A. K. Intracellular determinants of hippocampal CA1 place and silent cell activity in a novel environment. *Neuron* **70**, 109-120, doi:10.1016/j.neuron.2011.03.006 (2011).
- 20 Davoudi, H. & Foster, D. J. Acute silencing of hippocampal CA3 reveals a dominant role in place field responses. *Nat Neurosci* **22**, 337-342, doi:10.1038/s41593-018-0321-z (2019).
- 21 Mankin, E. A., Diehl, G. W., Sparks, F. T., Leutgeb, S. & Leutgeb, J. K. Hippocampal CA2 activity patterns change over time to a larger extent than between spatial contexts. *Neuron* **85**, 190-201, doi:10.1016/j.neuron.2014.12.001 (2015).
- 22 Hunsaker, M. R., Rosenberg, J. S. & Kesner, R. P. The role of the dentate gyrus, CA3a,b, and CA3c for detecting spatial and environmental novelty. *Hippocampus* **18**, 1064-1073, doi:10.1002/hipo.20464 (2008).
- 23 Dolleman-van der Weel, M. J. *et al.* The nucleus reuniens of the thalamus sits at the nexus of a hippocampus and medial prefrontal cortex circuit enabling memory and behavior. *Learn Mem* **26**, 191-205, doi:10.1101/lm.048389.118 (2019).
- 24 Cohen, J. D., Bolstad, M. & Lee, A. K. Experience-dependent shaping of hippocampal CA1 intracellular activity in novel and familiar environments. *Elife* **6**, doi:10.7554/eLife.23040 (2017).
- 25 Nitz, D. & McNaughton, B. Differential modulation of CA1 and dentate gyrus interneurons during exploration of novel environments. *J Neurophysiol* **91**, 863-872, doi:10.1152/jn.00614.2003 (2004).
- 26 Geiller, T., Fattahi, M., Choi, J. S. & Royer, S. Place cells are more strongly tied to landmarks in deep than in superficial CA1. *Nat Commun* **8**, 14531, doi:10.1038/ncomms14531 (2017).
- 27 Madar, A. D., Ewell, L. A. & Jones, M. V. Pattern separation of spiketrains in hippocampal neurons. *Scientific reports* **9**, 1-20 (2019).
- 28 Waller, L. A., Smith, D., Childs, J. E. & Real, L. A. Monte Carlo assessments of goodness-of-fit for ecological simulation models. *Ecological Modelling* **164**, 49-63 (2003).

Reviewers' Comments:

Reviewer #2:

Remarks to the Author:

The authors have addressed all my concerns analytically or conceptually. I have therefore no further requests.

Reviewer #3:

Remarks to the Author:

The manuscript has been improved and my concerns have been addressed.